

# SAGE III/ISS aerosol/cloud categorization and its impact on GloSSAC

Mahesh Kovilakam[1,2], Larry W. Thomason[2], and Travis Knepp[2]

[1]SSAI, Hampton, Virginia, USA
[2]NASA Langley Research Center, Hampton, Virginia, USA

**Correspondence:** Mahesh Kovilakam (mahesh.kovilakam@nasa.gov)

**Abstract.** The Stratospheric Aerosol and Gas Experiment on the International Space Station (SAGE III/ISS) began its mission in June 2017. SAGE III/ISS is an updated version of the SAGE III on Meteor (SAGE III/M3M) instrument that makes observations of stratospheric aerosol extinction coefficient at wavelengths that range from 385 to 1550 nm with a near global coverage between 60°S and 60°N. While SAGE III/ISS makes reliable and robust solar occultation measurements in stratosphere, similar to its predecessors, interpreting aerosol extinction measurements in the vicinity of tropopause and in the troposphere have been a challenge for all SAGE instruments. Herein, we discuss some of the challenges associated with discriminating between aerosols and clouds within the extinction measurements and describe the methods implemented to categorize clouds and aerosols using available SAGE III/ISS aerosol measurements. This cloud/aerosol categorization method is based on the results of Thomason and Vernier (2013) with some modifications that now incorporate the influence of recent volcanic/PyroCb events and a new method of locating aerosol centroid based on k-medoid clustering. We use version 5.2 of SAGE III/ISS extinction coefficients for the analysis. The current algorithm now classifies standard (background) and non-standard (enhanced) aerosols in the stratosphere and identify enhanced aerosols and aerosol/cloud mixture in the tropopause region. SAGE data is an important dataset in the GloSSAC data base and therefore, the impact of cloud-filtered aerosol extinction coefficient measurements on the latest version of GloSSAC (version 2.2) is also discussed.

## 1 Introduction

The Stratospheric Aerosol and Gas Experiment on the International Space Station (SAGE III/ISS) began collecting data in June 2017 and is an updated version of the SAGE III on Meteor (SAGE III/M3M) instrument. SAGE III/ISS works similar to its predecessors (e.g. Mauldin et al., 1985; Thomason et al., 2010), retrieving vertical profiles of multi-wavelength aerosol extinction coefficient (384, 449, 521, 602, 676, 756, 869, 1022, and 1544 nm) along with other gaseous measurements such as ozone ($O_3$), water vapor ($H_2O$), and Nitrogen dioxide ($NO_2$) using solar occultation technique. The SAGE family of instruments have been instrumental in providing vertical profiles of global stratospheric aerosol that have been used by various correlative measurements for comparison and validation purposes (e.g. Hervig and Deshler, 2002; Deshler et al., 2003, 2019; Rieger et al., 2019; Bourassa et al., 2019) . Further, the SAGE series of measurements have been used for providing a global space-based





stratospheric aerosol climatology (GloSSAC) with other space-based measurements (Thomason et al., 2018; Kovilakam et al., 25 2020).

Several studies have shown the importance of the impact of stratospheric aerosols on determining the energy balance of the atmosphere (e.g. Hofmann and Solomon, 1989; Fahey et al., 1993; Minnis et al., 1993). Recent years have witnessed frequent small to moderate volcanic eruptions and wildfire events that injected aerosols into the stratosphere, which resulted in radiative, chemical, and dynamical impact in the stratosphere (Peterson et al., 2018; Yu et al., 2019; Kablick III et al., 2020; 30 Knepp et al., 2021). Additionally, studies have shown that relatively low aerosol loading can also have radiative impact in the stratosphere (Vernier et al., 2011). It is therefore important to get accurate information of stratospheric aerosol extinction during and following such events and to have the ability to distinguish between aerosol associated with these events and clouds. The objective of this study was to develop an aerosol-cloud separation algorithm that enables this distinction under perturbed conditions following events such as volcanic eruptions and Pyrocumulonimbus (PyroCb) events.

Several cloud and aerosol discrimination studies have been documented in the past using SAGE data (e.g. Kent and Mc-Cormick, 1991; Kent et al., 1993; Wang et al., 1994; Kent et al., 1997a, b; Thomason and Vernier, 2013). These studies have one thing in common, which is the usage of multi-wavelength SAGE extinction coefficient measurements to infer information about particle sizes. Kent et al. (1993) developed a method to separate aerosol and clouds using an extinction coefficient distribution based on SAGE II measurements at 525 and 1020 nm. A key finding of Kent et al. (1993) was that optically thin clouds 40 are in fact aerosol-cloud mixtures and they concluded that the transition from aerosol to aerosol/cloud occurs over a continuum. Wang et al. (1994) later investigated aerosol-cloud interaction of tropical high clouds using the same wavelength combination as Kent et al. (1993), but with additional information of temperature to identify the presence of high clouds in the tropical region. A different approach using three wavelength combinations (525,1020, and 1550 nm) was later developed by Kent et al. (1997a) for SAGE III/M3M to identify cloud height. While studies show differences in identifying tropopause height between 45 different re-analyses (e.g. Boothe and Homeyer, 2017; Manney et al., 2017; Xian and Homeyer, 2019), accurate estimation of tropopause height is an important factor in identifying clouds in the vicinity of tropopause, so as to not bias aerosol data.

Herein, we describe a cloud screening algorithm for SAGE III/ISS to study the challenges in identifying pure aerosol and aerosol-cloud mixture from SAGE III/ISS observations and their impact on the development of GloSSAC version 2.2 (v2.2). It is worthwhile to note here that the aerosol record post-2017 witnessed several volcanic eruptions and wildfire events that 50 injected particles into the stratosphere, further complicating the separation of aerosol and clouds near the tropopause. As previously noted, Thomason and Vernier (2013) (hereafter TV13) have studied the challenges in separating aerosol and cloud using SAGE II observations particularly during a volcanically quiescent period for the tropical Upper Troposphere and Lower Stratosphere (UTLS) region. For SAGE III/ISS, it becomes even more challenging due to frequent moderate volcanic eruptions and wildfire events that occurred during SAGE III/ISS measurements. Here, we describe a modified version of TV13 to account 55 for the perturbed events, thereby implementing the method not only for the tropics but for the entire global dataset.



## 1.1 SAGE III/ISS muti-wavelength extinction measurements and version changes

A newer version (version 5.2) of SAGE III/ISS data is recently released. The changes occurred in version 5.2 are described in the version 5.2 release notes (https://sage.nasa.gov/wp-content/uploads/2021/07/SAGEIII_Release_Notes_v5.2.pdf). Some of the broad changes in the solar product in version 5.2 includes non-smoothing of solar data products, altitude registration cor-
rection and an automated "QA" process. While SAGE III/ISS aerosol extinction measurements have been used for validation, comparison and long-term climatology purposes (e.g. Bourassa et al., 2019; Rieger et al., 2019; Kar et al., 2019; Kovilakam et al., 2020), a negative bias in the aerosol channels ( 521, 602, and 676 nm) close to Chappius band has been recently noted (Wang et al., 2020). This reported bias is currently being investigated. SAGE III/ISS makes measurement similar to SAGE II up to a line-of-sight (LOS) optical depth close to 7. We, therefore follow the TV13 method to first terminate each profile at the
first altitude at which aerosol extinction exceeds $2 \times 10^{-2}$ km$^{-1}$ or the LOS optical depth exceeds 7.

For SAGE III/ISS version 5.2, we notice that some profiles report negative extinction coefficients in the UTLS. Due to the noise characteristics of the SAGE III/ISS transmission data, some SAGE III/ISS profiles have significant noise in them that can lead to product uncertainties that do not satisfactorily account for this negative retrieval. For example, Figure 1a demonstrates this phenomenon in the UTLS with the color-coded dots indicating the relative uncertainty (extinction coefficients were plotted
as absolute values to accommodate the log scale). Here, it was observed that two points had negative extinction coefficieints, but had relative uncertainties less than 50%. Normally, these negative values occur at higher altitudes, which is not unexpected, or have uncertainties that are large enough to make the extinction coefficients effectively indistinguishable from zero. However, that is not what was observed for the points in Figure 1a. This phenomenon seems to occur predominantly in the vicinity of a large positive extinction coefficient, particularly between 12 and 14 km in Figure 1. Because of the inherent sensitivity in
separating clouds and aerosols in occultation-type data, we developed a filtering algorithm to mitigate the influence of these negative extinction coefficients on subsequent analyses. This filtering algorithm scans for negative values from the top of the profile downward, starting at an altitude of 25 km down to where the profile terminates. For higher altitudes ($\geq 25$ km), negative values mostly occur due to noise and errors in the removal of ozone and molecular scattering, therefore higher altitude negative extinction coefficients are retained. However, when they occur at lower altitudes particularly in the vicinity of tropopause or
below is a sign that the measurements are less reliable. Therefore, the screening algorithm is hard coded to terminate profiles just above the altitude where the first negative value occurs in the troposphere. However, the algorithm is more selective on how data are removed within the stratosphere. For example, when a negative extinction coefficient was observed in the stratosphere the data points at adjacent altitudes (i.e., the altitude immediately above and below the negative value) were removed from the data set; resulting on only 3 data points being removed. However, when a negative extinction coefficient was observed in the
troposphere that data point as well as all data points below it were removed. This filtering mechanism can be seen at work in Figure 1b where three points were removed around 11 km and all data below 6.5 km were removed.



## 2 Aerosol and Cloud identification in SAGE III/ISS observations

Several previous studies (e.g. Thomason et al., 2008; Thomason and Vernier, 2013) have provided in depth analyses of multi-wavelength stratospheric aerosol extinction coefficients from SAGE II and how extinction ratios relate to the particle sizes on the basis of Mie theory. Theoretically, aerosol extinction efficiency at any wavelength can be computed from Mie theory using the underlying particle size distribution, particle composition through index of refraction, and shape of the distribution. Stratospheric aerosols are primarily sub-micron spherical liquid droplets that are typically composed of sulfuric acid and water (Rosen, 1971). Particle composition is estimated following Steele and Hamill (1981) with index of refraction from Palmer and Williams (1975). Particle size dependence of extinction efficiency at SAGE III/ISS channels is shown in Figure 2. Further, Figure 2b shows extinction ratios computed from extinction efficiency kernels at 525 and 1020 as well as 756 and 1544 nm that provide information on how extinction ratios are related to particle size. The extinction ratio between 525 and 1020 nm becomes unity as particle size approaches 0.5 μm, suggesting that it becomes difficult to distinguish aerosol from cloud as the average particle size gets to 0.5 μm, which typically occurs following large volcanic eruptions.

### 2.1 TV13 Method

In this section, we briefly discuss the method employed by TV13, as well as any specific differences between their method and ours. TV13 demonstrated how mutli-wavelength measurements can be used to separate aerosol and aerosol/cloud mixture during the stratospheric background period from 1999 through 2005. The approach used by TV13 was based on the 525:1020 nm aerosol extinction ratio. For their approach, as a first step, a distinction between primary aerosol and enhanced aerosol was made for the data with extinction ratios larger than 2 in order to avoid any possible cloud contamination (smaller extinction ratios with larger extinction coefficients represent larger particles such as clouds). A probability density function of extinction and a median absolute deviation statistic were used to distinguish between the primary aerosol centroid and enhanced aerosol. The separation between primary aerosol and enhanced aerosol ($k_0$) is computed using median absolute deviation statistics, which is defined as $k_0 = k_a + 3.0*MAD$, where $k_a$ is the median extinction coefficient of the distribution and MAD is the median absolute deviation. TV13 also noted that in the UTLS region, 95% of the data points with extinction ratios greater than 2 lie below the cutoff value $k_0$. For SAGE measurements, we interpret our data as mixtures of aerosol/cloud because the long path lengths through the atmosphere characteristic of SAGE-like observations see a combination of aerosol and cloud the expected outcome rather than a purely 'cloud' observation. TV13 used an empirical model based on aerosol centroid an artificial cloud centroid with extinction ratio of 1 and an extinction coefficient of $10^{-1}$ km$^{-1}$. The aerosol centroid co-ordinates are computed using the median of the distribution. Here, we reproduce this method with specific changes as we compare TV13 method with the new method in the following section.

The TV13 method was modified to facilitate comparison between the results of the TV13 method and the new method presented herein (section 2.2). One such change is in the wavelength combination used. While the most commonly used wavelength combinations for SAGE series of measurements are 525 and 1020 nm, we prefer to use a different wavelength combination of 756 and 1544 nm because 756 and 1544 is a wavelength range of a factor of 2, similar to 525 and 1020.





Moreover, unlike 525 nm, event termination is less likely to occur at 756 nm channel significantly above 1544 nm channel, particularly when large aerosols/clouds are encountered. Additionally, another reason to select 756:1544 wavelength combination is that the negative bias reported in the 525 nm channel in SAGE III/ISS (Wang et al., 2020). TV13 used SAGE II data collected between 1999 and 2005 and broke that dataset up on a seasonal basis. Here, in our analysis we use SAGE III/ISS data collected between 2017 and 2021. Instead of breaking up the dataset by season, we divide that data according to month

and used the 756:1544 nm extinction ratio for the reasons mentioned above.

Figure 3a depicts the 756 to 1544 nm extinction ratio as a function of 1544 nm extinction from SAGE III/ISS measurements for February during the period 2017 through 2021 at 15 km altitude.The vertical red line in Figure 3 shows $k_0$, which is computed in the same method as TV13. Here, we reproduce this method using SAGE III data in Figure 3a. Following TV13 method, we use an offset of 0.4 for the empirical model ratios (R) that accounts for the spread in the aerosol/cloud mixture tail

observed in Figure 3a, which is mostly a result of increased measurement noise in 756 nm extinction coefficient. The extinction measurements that are larger than $k_0$ but with extinction ratio less than R are considered as aerosol/cloud mixtures. It is also noted that there is a wedge-shaped region between $k_0$ and the modeled curve, which is denoted as "W" in Figure 3. In the TV13 method, data that fall in this region were counted as aerosol/cloud mixtures and were removed from further analyses. Figure 3b shows the data after filtering the tail of the distribution and the data that fall inside the wedge shaped area. The

TV13 method works reasonably well during background periods when there is no enhancement of aerosol so that particle size remains realtively small ($< 0.5$ μm) and there is no ambiguity in the relationship with extinction ratio and the particle size. However, the classification becomes challenging, following volcanic/PyroCb events where the extinction coefficients are enhanced with larger aerosol extinction for which the classification of the tail of the distribution is problematic as shown in Figure 3. In such cases, the enhanced aerosol could be mixed with aerosol/cloud mixture category as they are likely filtered

out from the aerosol category, which could potentially result in an underestimation of aerosol extinction coefficient. Such cases can frequently occur in the vicinity of the tropopause or in the lower stratosphere where many such enhanced aerosols could be mis-classified as aerosol/cloud mixtures. We also note that perturbed events can cause more than one aerosol centroid which is due to inter-hemispheric differences in the data following the events. In Figure 3, there appears to be two clusters in the background aerosol region, which is a result of perturbed events. In the following section, a modified approached of TV13 with

classifications is described.

## 2.2 Perturbed Stratosphere and New Method.

For SAGE III/ISS aerosol extinction, we have an advantage of additional wavelength measurements including a longer wavelength (1544 nm) that can be used for the analyses. As noted above, while SAGE III/ISS measurements have been used in many studies, due to a reported negative bias in the 525 nm channel (Wang et al., 2020), we now apply a simple correction by

spectrally interpolating extinction between 450 and 756 nm channel using Ångström exponent (Knepp et al., 2021). While the most commonly used wavelength combinations for SAGE series of measurements are 525 and 1020 nm, due to the reasons listed in section 2.1, we prefer to use a different wavelength combination of 756 and 1544. We, however compared 525 and





1020 nm wavelength combination for historical reasons and observe that both wavelength combinations (756 /1544 nm and 525 /1020 nm) yield similar results in the stratosphere (not shown here).

SAGE III/ISS data witnessed many small to moderate eruptions and several PyroCb events that reached the stratosphere. Table 1 lists those events with event date and latitude of occurrence. The frequency of occurrence of these events make it more challenging to employ a cloud screening algorithm for SAGE III/ISS. An enhancement of aerosol extinction following these events is therefore important as in some cases, enhanced aerosols could be large enough to be misinterpreted as clouds, particularly when they are mixed with aerosol/cloud mixture in the tail of the distribution shown in Figure 3a. Our goal here

is to first identify enhancement in aerosol extinction following these events and the time it takes to get back to the background level. We, therefore analyzed the data following these events to look for enhanced aerosol loading. We then employed an approach to look for any enhancement in the data based on median absolute deviation statistics to compute outliers in the data. The outliers for these events are in fact perturbed extinction coefficients. Median absolute deviation statistics is computed on a monthly basis with a 20 degree latitude band that is centered at the volcanic/fire event latitude. We, then define an outlier

extinction coefficient from the monthly probability density function distribution. The outlier extinction coefficient, $k_0 = k_a + 3.5*MAD$, where $k_a$ is the median extinction coefficient of the distribution and MAD is the median absolute deviation. We use a factor of 3.5 to estimate outlier based on modified Z-scores of a simulation study (Iglewicz and Hoaglin, 1993) that shows an absolute value of 3.5 can be used to define potential outliers in the above equation.

    Figure 4 shows a time series of $k_0$ for 1544 nm for various latitude bands for all events shown in Table 1. Based on the

170 $k_0$ time series, the timeframe of volcanic/fire event is determined. $k_0$ is an estimate of the extreme value that represents the enhancement of extinction coefficient due to any volcanic/fire event. For each event, the we track the enhancement in aerosol extinction following an event and the time it takes to get back to the background level using the red symbol shown in the time series plot. The altitudes are chosen in such a way that they represent average tropopause altitude for each latitude band. In the vicinity of tropopause, separating aerosols from clouds is always challenging, particularly following such events as it becomes

hard to distinguish between background as well as enhanced aerosol/clouds (Thomason and Vernier, 2013). We therefore make use of the timeframe from the time series plots that will be incorporated in the cloud algorithm so that we are able to track the event and retain the data in the vicinity of the tropopause with a different flag named " Volcanic Aerosol/Tropopause Cloud". A detailed description of flags in the data will be discussed in Section 2.2.2.

### 2.2.1    Identifying centroid of the distribution based on measurements.

The first step in the cloud-screening was to estimate the influence of perturbed events based on the time series plot for different latitude bands. While we follow the TV13 method, some modification to their work is needed in order to incorporate influence of volcanic/fire events. So, the next step is to locate the aerosol centroid from the extinction ratio versus extinction plot so that we could effectively implement the aerosol model used in TV13.

    For the TV13 method, we used median of the distribution of the global data on a monthly basis. However, for SAGE III/ISS,

we noticed that the perturbed events cause inter-hemispheric differences in the data, leading to more than one cluster in the data. Figure 5 shows scatter plots of extinction ratio versus extinction for two of the perturbed events. The upper panel of the



figure shows how extinction ratios (756/1544) change with respect to 1544 nm extinction, following Canadian Wildfire event that occurred in August 2017. While all of the data are plotted, we use different colors for different latitude bands to show the influence of any such event in the data. Please note that the data presented in Figure 5 was plotted at 2 altitudes: 11 km (panels a-d) and 17 km (panels e-h). The 11 km altitude was used because it is approximately representative of the average tropopause height for this latitude band and the 17 km altitude is chosen because it represents approximate tropopause altitude for the tropics. It is evident from the figure that there is a distinct enhancement of extinction coefficient in the northern latitude band (20°N-80°N) following the event while outlier statistics show that the data points fall beyond the $k_0$ value are clearly enhanced extinction from the perturbed event. Similarly, the lower panel shows data following the Ambae eruption in July 2018 and demonstrates distinctly enhanced extinction from the southern latitude band (80°S-20°S). Additionally, these global monthly data plots suggest there are inter-hemispheric differences in extinction coefficient.

Locating the aerosol centroid is challenging, particularly following a volcanic/fire event as it is evident from Figure 5 that there are more than one cluster with inter-hemispheric differences. Therefore, we employed an unsupervised machine learning clustering algorithm called as "k-medoid clustering" (e.g. Kaufman and Rousseeuw, 1990; Park and Jun, 2009). The K-medoid clustering algorithm is more robust in identifying noise and outliers as it picks one of the cluster members as the medoid (a medoid is a data point that has the shortest total distance to other members of the cluster). For the analyses here, we use two medoids that are based on the scatter plots shown in Figure 5 . We tested our algorithm on the global data and it works reasonably well in identifying the centroid in the extinction ratio plots shown in Figure 5 . While the algorithm works well in most cases, with the inter-hemispheric differences that we see in the data, we decided to implement the clustering algorithm for different latitude bands to avoid any bias in identifying centroids. We, therefore divided monthly data into two latitude bands: 1) 80°S-20°N, and 2) 20°N-80°N. Initially, the tropical latitude band (20°S-20°N) was used as a separate latitude band but lack of tropical data for statistical analysis forced us to combine the tropical latitude band to the southern latitude band (80°S-20°S). We, therefore perform aerosol categorization based on these two latitude bands (80°S-20°N and 20°N-80°N), which will be described in the following section.

### 2.2.2 Categorizing aerosols and aerosols-cloud mixtures

While we follow the method used in TV13, a modified approach is used here as SAGE III/ISS data record has several low to moderate volcanic eruptions and some extreme fire events which makes separation of aerosol from aerosol-cloud mixtures challenging. In Figure  6 , we show how aerosol/categorization is done by showing examples of two perturbed events: (a) Canadian Wildfire and (b) Ambae eruption that occurred at 51°N and and 15°S respectively. In our revised method, we use monthly median absolute statistics to estimate the outlier extinction coefficient ($k_0$). The vertical red solid line in each plot represents $k_0$, whereas the horizontal red lines indicate the location of a hypothetical aerosol/cloud mixture's extinction ratio of 1 with a fixed offset of 0.4 similar to TV13. We therefore divide the plotting area into four quadrants based on $k_0$ and an extinction ratio of 1.4. The upper left quadrant is identified as "background" aerosol, while data in the upper right quadrant represent "perturbed" aerosol. However, the data points that fall in the lower two quadrants are difficult to categorize. Since the extinction ratio is less than 1.4 and extinction value greater than $k_0$ then those particles are large enough to be misinterpreted as





clouds, following a volcanic/fire event. We, therefore implemented a different method that dictates how flags are determined in the lower two quadrants. We use time frame from the time series analysis shown in Figure 4 for each event and for latitude band within 20° of the latitude of occurrence of any perturbed event. The influence of the perturbed event on extinction is decided on whether an extinction coefficient data point falls within the prescribed latitude band which is based on a perturbed event

and whether the extinction ratio falls below 1.4. While, TV13 categorized this lower tail of the distribution as "aerosol/cloud mixture", we make use of time and latitude of the event along with tropopause height to incorporate the influence of any perturbed event. Based on the time frame (the red symbols in Figure 4) after each event, we additionally use a tropopause height data as the lower altitude range for which this method is valid. For the circumstances of a perturbed event, we then relax the "Aerosol/Cloud Mixture" category and flag them as "Enhanced aerosol/Tropopause cloud" based on the above criteria,

whereas the data that do not fit the above criteria is flagged as "Aerosol/Cloud Mixture". By doing this, we retain some of the near tropopause data that may have been influenced by such perturbed events that were otherwise classified as "aerosol/cloud mixture" in TV13 method. The "enhanced aerosol/tropopause cloud" flag is now added to the aerosol category for all our analyses. The viewing geometry of SAGE III/ISS is such that separating enhanced aerosols from clouds in the vicinity of UTLS is otherwise difficult.

Figure 6 represents the revised aerosol/cloud categorization. The upper panel in Figure 6 shows scatter plots of 756/1544 extinction ratio versus 1544 extinction for 20°N-80°N following the Canadian Wildfire event in 2017. It is clearly evident that an enhancement in extinction coefficient occurs following the event particularly in the 20°N-80°N latitude band with aerosol centroid coordinates clearly suggesting an increase in 1544 nm extinction and a decrease in the extinction ratio. We use different flags to identify different types of aerosols as mentioned above. The plots shown in Figure 6 are for an altitude of 11 km and

17 km , which is roughly the tropopause altitude for the latitudes where Canadian Wildfires and Ambae eruption occurred. So, with the criteria used above, we classify the near tropopause aerosol following any event within a latitude band of 20 degree centered at the event latitude as "Enhanced Aerosol/Tropopause Cloud" (Green symbols in Figure 6). The "Enhanced Aerosol/Tropopause Cloud" flag is added into the aerosol category in all the analyses. This may cause an enhancement in aerosol extinction coefficient but we retain these data points because they might contain important information on enhanced

aerosols from the perturbed event. An empirical model is fit to the observations following Equation 4 of TV13, which is shown in Figure 6. The model is computed using information on aerosol and cloud centroid co-ordinates. TV13 use median of the distribution to locate aerosol centroid whereas the current method used "k-medoid clustering" described in section 2.2.2 to locate aerosol centroid. The cloud centroid co-ordinates are empirically determined as 1544 nm extinction and 756 to 1544 nm extinction ratio being set to $1.0 X 10^{-1}$ km $^{-1}$ and 1.4 respectively, similar to TV13 method.

## 2.3 Comparison between TV13 and New Method

We compare TV13 method and the new method after filtering out possible aerosol/cloud mixture. Figure 7 shows scatter plot of 756 to 1544 nm extinction ratios as a function of 1544 nm extinction at two different altitudes. The data shown in Figure 7 is for the period 2017-2021 for their respective altitudes. Figure 7 depicts the entire SAGE III/ISS data at 11 and 17 km. The noted difference in the new method is that the retention of some of the data in the tail of the distribution between extinction ratio





of 1.4 and 2.0, whereas TV13 method removes the data in this region as it is evident in Figure 7a,c. TV13 method, however retains data that fall in the lower left quadrant where extinction ratios are less than 2.0 and for extinction less than $k_0$, whereas in the new method these data have been removed as these data with smaller extinction ratios and relatively smaller extinction coefficients are possibly of larger extinction uncertainty and are less reliable. For the latitudes $> 55°$, we make an effort to identify polar stratospheric clouds (PSC) in the new method by applying a temperature based filter. The PSCs are filtered out

when the ambient temperature falls below 200K.

## 3   Implications of cloud-screening on GloSSAC

Stratospheric aerosol is an important component in determining the radiative and chemical balance of the atmosphere. Many Global Climate Models (GCMs) do not have an interactive aerosol module to treat stratospheric aerosol and therefore depend on global measurements on a long-term basis. Therefore, Global Space-based Stratospheric Aerosol Climatology (GloSSAC) was

created first in 2018 (Thomason et al., 2018) to support the climate modeling community for Couple Model Intercomparison Project Phase 6 (CMIP6) project (Eyring et al., 2016). For GloSSAC, the SAGE series of measurements play a vital role in the long-term data starting from 1979 through present, excluding post-SAGE II era (August 2005-May 2017) during which other space-based measurements were used to fill the gap (e.g. Thomason et al., 2018; Kovilakam et al., 2020). While other measurements were available to fill the gap for the post-SAGE II era, an important factor missing in those measurements

were measurements of extinction coefficient at multiple wavelengths. For GloSSAC version 2.0 (Kovilakam et al., 2020), we extended the data set to December 2018 with the inclusion of SAGE III/ISS multiple wavelength data from June 2017. It is known that the presence of cloud in the vicinity of the UTLS makes it challenging for SAGE measurements to identify and separate pure aerosol from aerosol cloud mixture/enhanced aerosols, particularly following a perturbed event such as volcanic eruption or PyroCb as removing those data points (i.e. lower extinction ratios with higher extinction coefficients) could lead

to an underestimation of aerosol extinction in the UTLS region. Recent changes in the stratospheric aerosol loading in the UTLS region have received significant attention in the scientific community (e.g. Bourassa et al., 2012; Vernier et al., 2015) as increased aerosol loading in this region can have larger impact on radiative and chemical balance. Therefore, it is important to identify aerosols more accurately in the vicinity of tropopause particularly following events that perturb the stratosphere.

For GloSSAC version 2.0 (v 2.0), SAGE III/ISS aerosol extinction coefficient data has been incorporated with a simple

extinction ratio based cloud filter approach to avoid any possible cloud contamination in the aerosol extinction data. By using this simple method, some enhanced aerosols from any perturbed event (e.g. volcanic eruptions, PyroCb events) may have been mistakenly flagged as clouds during SAGE III/ISS measurements, particularly in the vicinity of tropopause and lower stratosphere. It is therefore important to address this issue in the aerosol data, particularly when it is used for a long-term climatology such as GloSSAC. Therefore, we incorporate the revised cloud screening method described in section 2.0 into the

GloSSAC version 2.2 (v 2.2).

Here, SAGE III/ISS data is zonally averaged into 5°latitude bins, and 0.5 km altitude resolution on a monthly basis and incorporated into GloSSAC . Additionally, for this version (v2.2), we perform a linear interpolation along time axis to fill in





missing values at higher latitudes. For a future release, we plan to implement a reconstruction method for SAGE III/ISS to fill in missing data— a method similar to the one used for SAGE II in GloSSAC version 1.0 (Thomason et al., 2018). It should also be noted that we now use version 5.2 SAGE III/ISS products in GloSSAC (v2.2) with revised cloud screening algorithm, whereas SAGE III/ISS version 5.1 was used in GloSSAC (v2.0) (Kovilakam et al., 2020). Figure 8 shows the impact of revised cloud screening algorithm on SAGE III/ISS aerosol data. For the cloud screened product, we use three flags from the cloud screen algorithm, which are "Standard aerosol", "Perturbed aerosol", and "Enhanced Aerosol/Tropopause Cloud" respectively. We also note that the usage of "Enhanced Aerosol/Tropopause Cloud" flag as aerosol in the cloud screened product may introduce a positive bias in aerosol extinction, particularly in the vicinity of tropopause where separating aerosol from aerosol/cloud mixture is challenging. However, by using the timeframe shown in the monthly time series of $k_0$ in Figure 4 could alleviate the bias to some extent. Figure 8a,b shows zonally averaged altitude versus latitude plots of extinction coefficients at 525 nm for version 5.1 and 5.2 respectively. The impact of cloud screening is evident from Figure 8b with a clear enhancement in extinction coefficient in the latitudes $> 40°N$, particularly in the lower stratosphere. The enhanced extinction in version 5.2 is further evident from Figure 8c, which shows the ratio of extinction coefficient between version 5.1 and 5.2 . The ratio lower than 1 in Figure 8c suggests enhanced extinction coefficient in version 5.2, which occurs due to the retention of data in the vicinity of the tropopause, whereas in version 5.1 these data points were removed due to the usage of a simple extinction ratio filter.

## 3.1 Revisiting GloSSAC

A detailed description of the various measurements used in constructing GloSSAC (v 2.0) is shown in Figure 1 of Kovilakam et al. (2020). An interim version of GloSSAC was recently released (v 2.1) for which the new aerosol/cloud categorization described in section 2.2 was implemented, without initial filtering of spurious negative values in the events as described in section 1.1. In version 2.1 of GloSSAC, the data were extended to through 2020. We now extend data through 2021 as data from individual measurements become available for the year 2021. However, for the current version (v 2.2), there is no change in the individual measurements used but there are version changes in each data set in addition to the cloud-screening of SAGE III/ISS data as described above. The version changes of individual data sets are applicable only for the post-SAGE II era (September 2005- present) that now include a version change in SAGE III/ISS data as described in Section 1.1 with version changes in OSIRIS and CALIOP as described below.

We now use OSIRIS version 7.2 data compared to version 7.0 used in GloSSAC (v 2.0). For OSIRIS version 7.2, the background atmosphere was changed from ERA-interim to MERRA2 re-analysis for consistency, and failures and missing data were also fixed so that there are now more scans in general. Overall, there are no significant differences between version 7.0 and 7.2, particularly above the UTLS region. Due to a decline in the coverage of the instrument, data for the month of June of 2018 through 2021 are now absent from the entire data set. Please note that due to inclusion of additional scans, the version 7.2 data now includes increased number of profiles compared to version 7.0, which also results in differences between version 7.0 and 7.2.These differences are apparent, particularly in the high latitude bands. Additionally, in version 7.2, $NO_2$ regularization and ozone cross-section have been changed (personal communication, Adam Bourassa).





CALIPSO aerosol backscatter measurements have been valuable for GloSSAC, particularly in filling the gaps in the measurements, mostly in the polar latitudes. We use CALIPSO's Cloud-Aerosol Lidar with Orthogonal Polarization (CALIOP) aerosol backscatter coefficient data, which has also undergone a minor change to version 1.01 from July 2020, which is due to an upgrade required to the operating system on the production cluster (https://www-calipso.larc.nasa.gov/resources/calipso_users_guide/data_quality/level_all_v001_20201002.php) . For the year 2021, CALIOP data were available only till October 2021. We, therefore used only the data that were available at time of the analysis.

Here, we follow the methods described in Kovilakam et al. (2020) for merging OSIRIS, CALIOP, and SAGE III/ISS into GloSSAC data set. For this version (v2.2) of GloSSAC, we update the dataset post -2005 for which there are version changes on all the three individual data sets that are used (i.e. OSIRIS, CALIOP, and SAGE III/ISS). While the measurements of OSIRIS and CALIOP provide most of the data in GloSSAC for the period from 2005 through mid 2017, it may also be noted that there are changes in instruments and fundamental change in the measurements as noted previously (e.g. Thomason et al., 2018; Kovilakam et al., 2020). While OSIRIS and CALIOP instruments use a less direct technique compared to solar occultation, these instruments provide greatest density of measurements. However, OSIRIS and CALIOP have challenges in retrieving aerosol properties. For OSIRIS, the retrieved aerosol extinction at 750 nm depends on aerosol scattering phase function estimation which is related to aerosol size distribution and composition. CALIOP's primary measurement is backscatter coefficient which is measured at 532 nm. While CALIOP provides high density measurements even in the polar latitudes, there appears to have poor precision on individual measurements in the stratosphere. Therefore, we use averaging of individual measurements to provide a precise product comparable to those provided by OSIRIS or SAGE when incorporating the data set into GloSSAC. Since the GloSSAC aerosol extinction coefficients are provided at 525 and 1020 nm, the conversion from the CALIOP backscatter coefficient to extinction coefficient is another source of bias. A detailed description on biases related to individual instruments and their possible corrections is provided in Kovilakam et al. (2020).

## 3.2 Comparison of SAGE III/ISS data with OSIRIS

The primary wavelength at which OSIRIS extinction coefficient is retrieved is 750 nm. We, therefore need to convert the 750 nm extinction to the GloSSAC wavelengths which are at 525 and 1020 nm. Generally, the conversion to 525 nm is made using a constant Ångström exponent of 2.33 as noted in Rieger et al. (2015). While the comparison between OSIRIS and SAGE measurements are broadly in agreement, there appears to have been an overestimation of OSIRIS extinction in the lower stratosphere when compared against SAGE II and SAGE III/ISS (e.g. Rieger et al., 2015; Thomason et al., 2018; Kovilakam et al., 2020). Therefore, to maintain a long-term consistency between the data sets, it effectively requires that we bring OSIRIS in agreement with SAGE measurements. Following Kovilakam et al. (2020), a conformance process is performed to mitigate the differences between OSIRIS and SAGE measurements using a monthly climatology of pseudo Ångström exponent. A detailed description of this method is available in Kovilakam et al. (2020). While we follow the method provided in Kovilakam et al. (2020) for conforming OSIRIS data using pseudo Ångström monthly climatology, it is to be noted that we now include additional measurements available from OSIRIS and SAGE III/ISS from 2018 through 2020 to compute the monthly Ångström climatology. As noted in section 3.1, there are version changes in both the data sets that now introduce differences in the





Ångström climatology, which is used for the conformance process. Figure 9 shows the pseudo Ångström exponent monthly climatology that are computed using 750 nm OSIRIS and 525 nm SAGE II and SAGE III/ISS data. The difference between Figure 9 and Figure 7 of Kovilakam et al. (2020) arises mostly due to the relatively less comparable data between OSIRIS and SAGE III/ISS for GloSSAC version 2.0 where OSIRIS did not have data processed for the year 2018. Here, we now have
360 additional data available from January 2018 through December 2021 from both OSIRIS and SAGE III/ISS and it should also be noted that both data sets have undergone small version changes which may have contributed toward the differences in Figure 9 in comparison with Figure 7 of Kovilakam et al. (2020).

  Figure 10 shows a comparison measurement between OSIRIS and SAGE III/ISS for June 2017. For Figure 10a, the OSIRIS extinction is computed using a constant Ångström exponent of 2.33, whereas in Figure 10b the measurement is compared for
365 750 nm SAGE III/ISS measurement. Figure 10a,b show a reasonable agreement between OSIRIS and SAGE III/ISS except in the lower stratosphere and in the tropics where the percent difference exceeds 20 %. While Figure 10a shows same patterns as in Figure 10b, the differences are significantly larger in comparison with Figure 10b which suggests either a deficiency in the conversion process of OSIRIS extinction from 750 to 525 or SAGE III/ISS is biased low in the lower and middle stratosphere. Overall, the comparison of OSIRIS shows reasonable agreement except in the lower stratosphere where OSIRIS data appears
370 to have a high bias relative to SAGE III/ISS measurements. Therefore, to maintain long-term consistency between data sets, a conformance process is performed using a monthly climatology of pseudo Ångström exponent to convert OSIRIS extinction from 750 to 525 nm. The conformance process is detailed in section 2.4 of Kovilakam et al. (2020). Figure 10c shows the comparison between bias corrected OSIRIS and SAGE III/ISS measurements. Figure 10c shows a significant improvement of OSIRIS extinction coefficient in the lower stratosphere where the differences are significantly reduced in comparison with
375 Figure 10a. While this agreement looks better in Figure 10c for which the stratosphere is relatively less perturbed, there are some caveats as the usage of monthly climatology of pseudo Ångström exponent will be significantly different than the observed Ångström exponents, particularly following a perturbed event such as volcanic eruption or wildfire events. This becomes an issue for the period between SAGE II and SAGE III/ISS where we do not have any mutli-wavelength measurements during which many small to moderate volcanic eruptions occurred.

380  Recently, it is shown that many small to moderate eruptions can manifest themselves during SAGE II and III/ISS data record (Thomason et al., 2021). The size information inferred from 525 to 1020 nm extinction ratios show that a decrease in extinction ratio (increase in aerosol size) following large volcanic eruptions, whereas for small to moderate eruptions, extinction ratios are apparently slightly higher (smaller aerosol size) (Thomason et al., 2021). Therefore, we note that inferring aerosol size information for the post-SAGE II period (September 2005 through May 2017) is deficient, particularly following several small
385 to moderate volcanic eruptions. We use a similar conversion process for converting OSIRIS extinction coefficient at 750 nm to 1020 as described in section 2.4 of Kovilakam et al. (2020). While this conversion process is a better step forward in combining data sets into a uniform data set, we note that the pseudo Ångström exponent used for conformance does not have any physical meaning.





### 3.3 Comparison of SAGE III/ISS data with CALIOP and OSIRIS

Here, we follow the same method used in Kovilakam et al. (2020) to incorporate CALIOP data into GloSSAC data set. While CALIOP uses lidar to measure the aerosol backscatter coefficient at 532 nm, a source of bias occurs when backscatter coefficient is converted to extinction coefficient as the conversion process requires information of unknown aerosol composition and size distribution (Kar et al., 2019). Here, we use the CALIOP standard aerosol data product (Kar et al., 2019), with a minor version change from June 2020. For CALIOP data processing, a constant aerosol extinction-to-backscatter ratio of 50 sr is used

(Kar et al., 2019). CALIOP standard aerosol extinction is reported at 532 nm. Therefore, a constant Ångström exponent of 2.33 is used to convert extinction coefficient to 525 nm. Figure 11a,b show percent differences between standard CALIOP extinction coefficient and bias corrected OSIRIS and SAGE III/ISS extinction coefficient at 525 nm for November 2017. CALIOP is in reasonable agreement with OSIRIS and SAGE III/ISS except in the lower stratosphere and at higher latitudes ($> 50°$) where the differences are larger than 50 %. While we can attribute some of these differences to PyroCb event associated with

Canadian wildfire (Peterson et al., 2018), we note that similar differences persist even when the stratosphere is in the quiescent state. We, therefore use a conformance method described in Kovilakam et al. (2020) to reduce the bias between measurements. Following Kovilakam et al. (2020), we implement an empirical scale factor (SF) which is computed as the ratio of bias corrected OSIRIS extinction at 525 m to CALIOP backscatter coefficient at 532 nm. As noted in Kovilakam et al. (2020), we re-derive the backscatter using attenuated scattering ratio and molecular backscatter due to the fact that the standard aerosol

backscatter coefficient is retrieved using a lidar ratio of 50 sr (Kar et al., 2019). For GloSSAC (v2.2), we use this alternate backscatter coefficient same as the method used in GloSSAC v2.0.

Figure 12a depicts that annual median of SF on an altitude versus latitude basis. Figure 12a suggests that the SF values range from 10 at polar latitudes to about 65 in the tropical high altitudes. While SF in Figure 12a is in reasonable agreement with Figure 9 of Kovilakam et al. (2020), the differences in Figure 12 can be attributed to version changes and the additional

410 measurements available from 2018 through 2021. Figure 12b shows the relative standard deviation for SF in Figure 12a that shows SF is reasonably consistent except at polar latitudes where relative standard deviations are larger than 50 %. To compute the annual median of SF, we use data from 2006 through 2020 when both measurements are available on a monthly basis. We, then apply the conversion factors shown in Figure 12a to the entire CALIOP data set on an altitude versus latitude basis. This empirically scaled CALIOP 525 nm data is used for computing the difference plots shown in Figure 11c,d. It is evident

from these plots that the differences between the data sets is reduced and is mostly within $\pm$ 20 % when compared against Figure 11a,b for which the differences was $\geq$ 50 %. While the discrepancies between the data sets are reduced, they are not completely eliminated. We follow the same approach for converting CALIOP backscatter from 532 nm to 1020 nm extinction (not shown here). We plan to revise this method in a future version of GloSSAC as a time dependent SF can possibly be introduced after filling in missing values of OSIRIS and CALIOP monthly data using equivalent latitude.



### 3.4 Comparison of GloSSAC version 2.2 with version 2.0

To construct the GloSSAC data set, all individual measurements are gridded to the GloSSAC resolution ( monthly, 0.5 km altitude and 5 degree latitude resolution). As previously done for GloSSAC v2.0, from June 2017, we prioritize SAGE III/ISS data over OSIRIS and CALIOP. For the post-2017 data, several small to moderate volcanic events and a few large wildfire events have been reported (Table 1). It is therefore important to compare the differences between version 2.2 and version 2.0 GloSSAC data set. Figure 13 shows extinction coefficient for September 2017, following Canadian wildfire. Figure 13 a shows the GloSSAC (v 2.0) 525 nm extinction coefficient for September 2017, while Figure 13 b shows extinction coefficient for GloSSAC version 2.2. The ratios between Figure 13 a, and b are shown in Figure 13c. It is clearly evident from Figure 13 c that the revised cloud screen method used in v2.2 that apparently retains extinction data in the lower stratosphere that were otherwise removed in version 2.0 because of a simple extinction ratio filter— thereby enhancing aerosol extinction in version 2.2 for the latitude band between 35 and 50°N. The differences in the polar latitudes (> 60°) in version 2.2 could be attributed to changes occurred in individual data sets in version 2.2 as shown in Figure 14. Additionally, the differences in the polar latitudes could be a result of a linear interpolation scheme in time performed for SAGE III/ISS data in GloSSAC v2.2.

### 4 Stratospheric Aerosol Optical Depth

Stratospheric aerosol optical depth (SAOD) was incorporated as a separate variable in all previous versions of GloSSAC and therefore, we incorporate SAOD into GloSSAC version 2.2. Figure 15 shows monthly latitude versus time of SAOD for GloSSAC version 2.0 and version 2.2. While the data in version 2.0 and 2.2 remains the same for the period prior to 2005, there are differences between the versions for the post-2005 time period. It is worthwhile to note here that OSIRIS data used in version 2.2 has undergone minor changes with additional measurements included that affects the data broadly but more so in the polar latitudes. This may have caused differences for 525 nm SAOD in version 2.2 of GloSSAC, particularly for the southern hemispheric polar latitudes, which is evident in Figure 15c. Figure 15b,e show additional data from 2018 through 2021 in version 2.2, suggesting increased stratospheric aerosol loading due to several volcanic eruptions and wildfire events that are listed in Table 1. Globally averaged SAOD shows the differences between version 2.0 and 2.2 are generally within 20 % (Figure 16). Additionally, we note the impact of SAGE III/ISS cloud screening on GloSSAC version 2.2, which now shows an enhancement of aerosol extinction in comparison to version 2.0 following Canadian Wildfire event (July 2017) and Ambae volcanic eruption (July 2018). This is clearly evident in the percent difference plots shown in Figure 15 and 16 at both 525 and 1020 nm channels respectively.

### 5 Conclusions

We developed a revised method to categorize aerosol and clouds using SAGE III/ISS measurements. The primary goal behind a revised cloud filtering method was to account for the influence of recent volcanic eruptions and PyroCb events on stratospheric aerosol loading in SAGE III/ISS measurements. The revised method works reasonably well for the periods during and following





perturbed events such as volcanic/PyroCb. The influence of any perturbed activity in the stratosphere is estimated from the monthly time series of $k_0$, which is computed using median absolute deviation statistics and is now incorporated in the algorithm so that analyses that fall in this time frame is considered as "Perturbed" due to enhancement in $k_0$ value when compared against the "background" aerosol. Additionally, we use temperature based tropopause to classify the aerosols that are present in the
vicinity of the tropopause which is otherwise flagged as "aerosol/cloud mixture".

The implications of the revised cloud screen algorithm on GloSSAC data is also described. While there is no difference in the data prior to September 2005, the post-SAGE II era (September 2005-May 2017) clearly suggests differences between GloSSAC v 2.0 and this version (v 2.2). The differences between v2.0 and 2.2 are mostly attributable to version changes in the individual data sets, and revised cloud screening method used for SAGE III/ISS in v 2.2 of GloSSAC. While all individual
data sets for post-SAGE II era underwent version changes, the changes to OSIRIS is perceptible due to increased number of measurements in the latest version of OSIRIS (v7.1) which causes differences in the zonally averaged data for GloSSAC v2.2. While the differences are relatively low ($\leq 20\%$) except for the polar latitude for the period between post-SAGE II era and SAGE III/ISS era (June 2017- present), the difference between v2.0 and 2.2 of GloSSAC is relatively larger, particularly in the lower stratosphere following any perturbed events for the SAGE III/ISS time period (June 2017-present) which is attributable to
revised cloud screen algorithm that now retains data in the vicinity of the tropopause that was otherwise omitted with a simple extinction ratio based cloud screening in GloSSAC v2.0. The differences at the polar latitudes could be attributed to both version changes in individual data sets and the time interpolation of SAGE III/ISS data that is now implemented in version 2.2 due to discrepancies between data sets at the polar latitudes. While the same is true for SAOD where the time series of SAOD clearly shows the enhancement of SAOD following perturbed events, it is also noticeable that the enhancement of SAOD in
the southern polar latitude that was present in all previous versions of GloSSAC is now diminished, although not completely. The improvements in SAOD in the polar latitudes could be attributable to increased number of measurements available in the latest version of OSIRIS (v7.2) and thereby improved zonal averaging for those latitude bands. Additionally, in GloSSAC v2.2, a time interpolation of SAGE III/ISS data is now implemented which may have caused some differences at the higher latitude as well. We also note that there are slight differences between the interim version 2.1 and this version (v 2.2) of GloSSAC (not
shown here), as our aerosol/cloud categorization in both the versions remain relatively the same except that in version 2.2, an initial filtering of spurious negative values in the SAGE III/ISS events is implemented as described in section 1.1. Additionally, in GloSSAC 2.2, OSIRIS version changes from 7.1 to 7.2.

While there are noticeable improvements in GloSSAC v2.2, we plan to implement some changes in future that are listed below.

– We plan to revisit the way smoke events are represented in GloSSAC during SAGE II era. We plan to consider the possibility of not using any cloud clearing for SAGE II data sets just above the tropopause except in seasons with PSCs. It is likely that the current method is removing smoke aerosol data from SAGE II in the lower stratosphere due to a mix up with clouds particularly in the vicinity of the tropopause. We are currently revisiting this method to identify smoke events for SAGE II. Additionally, we plan to revisit the filling of missing data used during Pinatubo time period, and



the usage of Cryogenic Limb Array Etalon Spectrometer (CLAES) data, which is presumably underestimating aerosol extinction and thereby SAOD during and following Pinatubo time period.

  – We plan to include an improved scale factor for OSIRIS extinction to CALIOP backscatter ratios, and estimation of Ångström exponent from OSIRIS and SAGE III/ISS to convert OSIRIS 750 nm extinction to 525 and 1020 nm. Despite improvements in the data for the post-SAGE II era in GloSSAC across the versions, we understand the limitations of
the conversion method used particularly during periods when the stratosphere is perturbed due to volcanic and PyroCb activities. For CALIOP extinction estimation at 525 and 1020 nm from backscatter coefficient at 532 nm, we plan to implement a time dependent scale factor that will be computed after filling in OSIRIS and CALIOP missing values at higher latitudes with equivalent latitude approach which was implemented for the SAGE II data in GloSSAC. A similar equivalent latitude approach can be implemented for SAGE III/ISS data that will improve estimation of Ångström
exponent on a monthly basis which could then be used to convert OSIRIS 750 nm extinction to 525 and 1020 nm for the post-2017 data set.

*Data availability.*  The GloSSAC v2.2 netCDF file is available from the NASA Atmospheric Data Center (https://asdc.larc.nasa.gov/data/GloSSAC/GloSSAC_V2.2.nc) (NASA/LARC/SD/ASDC, 2022). The SAGE III/ISS and CALIOP data used in this study are available from NASA Atmospheric Data Center, while OSIRIS version 7.2 data are downloaded from https://arg.usask.ca/docs/osiris_v7.

*Author contributions.*  MK and LWT developed the idea and methodology used in this paper. MK carried out the analysis, while TK participated in the scientific discussion. MK wrote the manuscript, while all authors reviewed the manuscript and provided advice on the manuscript and figures.

*Competing interests.*  The authors declare that they have no conflict of interest.

*Acknowledgements.*  We acknowledge the support of NASA Science Mission Directorate and the SAGE II and III/ISS mission teams. The
SAGE mission is supported by the NASA Science Mission Directorate. SSAI personnel are supported through the STARSS III contract.



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

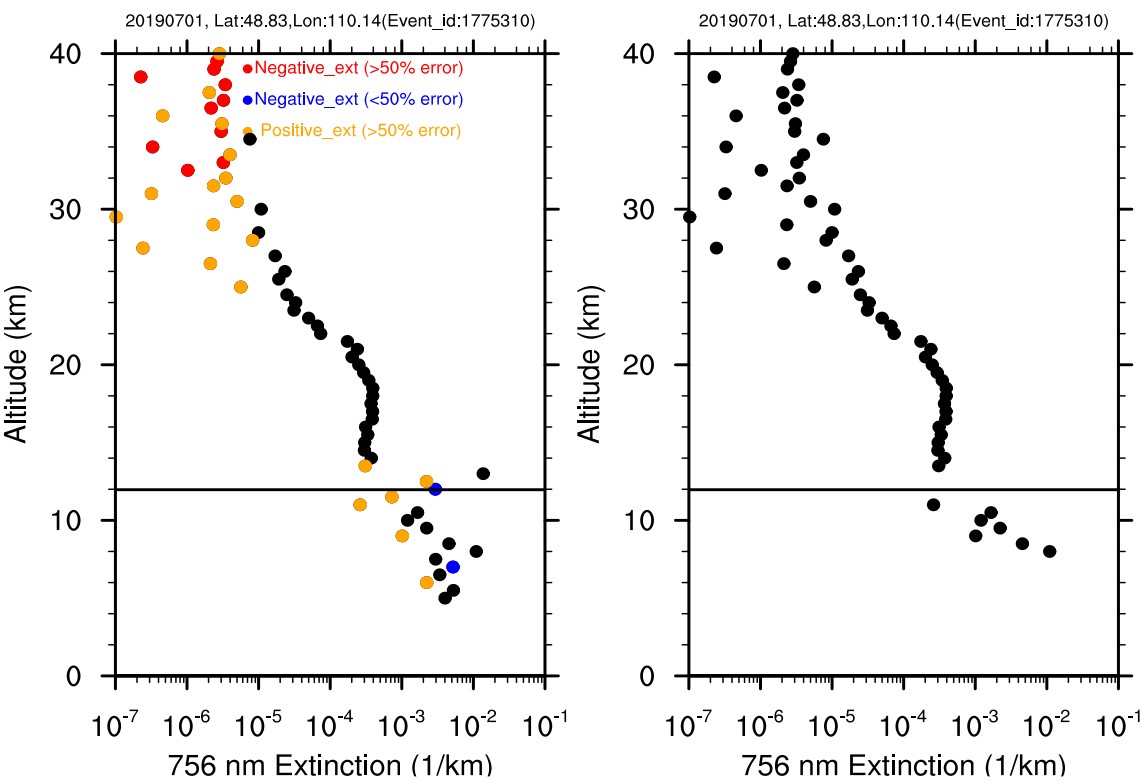

**Figure 1.** A sample extinction profile at 756 nm that shows how negative extinction values in the lower stratosphere as well as in the troposphere (Left). The right panel shows the extinction profile after filtering negative values. Red, and blue symbols in the left panel shows negative extinction values with uncertainty $> 50\%$ and $< 50\%$ respectively, while orange symbols represent positive extinction with $> 50\%$ uncertainty. The absolute value of the negative extinction coefficients (blue and red dots) are plotted to accommodate the log scale.

**Table 1.** Volcanic and PyroCb events used in this study

| Event Name | Event Date | Latitude |
|---|---|---|
| Canadian Wildfire (Cw) | 17 July 2017 | 51N |
| Ambae Eruption (Am) | 28 July 2018 | 15S |
| Ulawun Eruption (Ul) | 22 June 2019 | 5S |
| Raikoke Eruption (Ra) | 03 August 2019 | 48N |
| Australian Wildfire (Aw) | 06 January 2020 | 34S |
| California Creek Fire (Cc) | 01 September 2020 | 37N |
| La Soufriere (La) | 22 April 2021 | 13N |
| McKay Creek Fire (Mc) | 29 June 2021 | 54N |





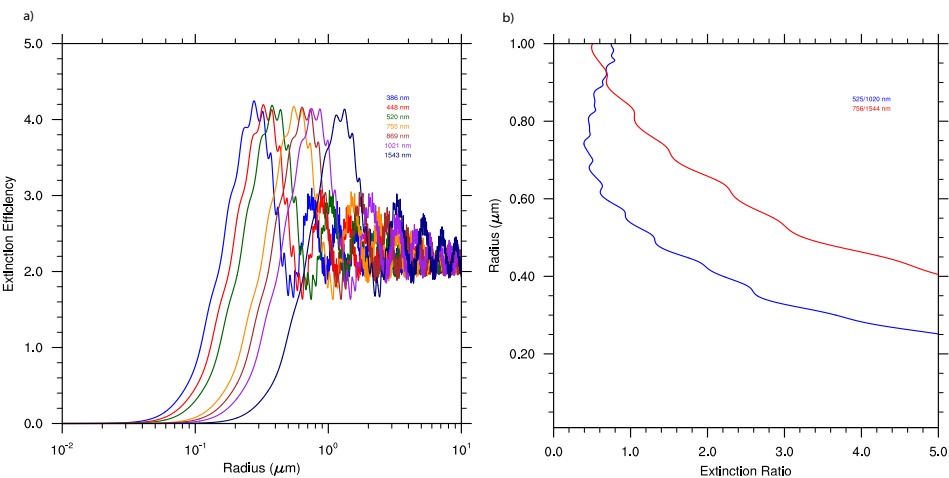

**Figure 2.** Extinction efficiency as a function radius for all SAGE III/ISS wavelengths (a) and radius versus extinction ratios (b).

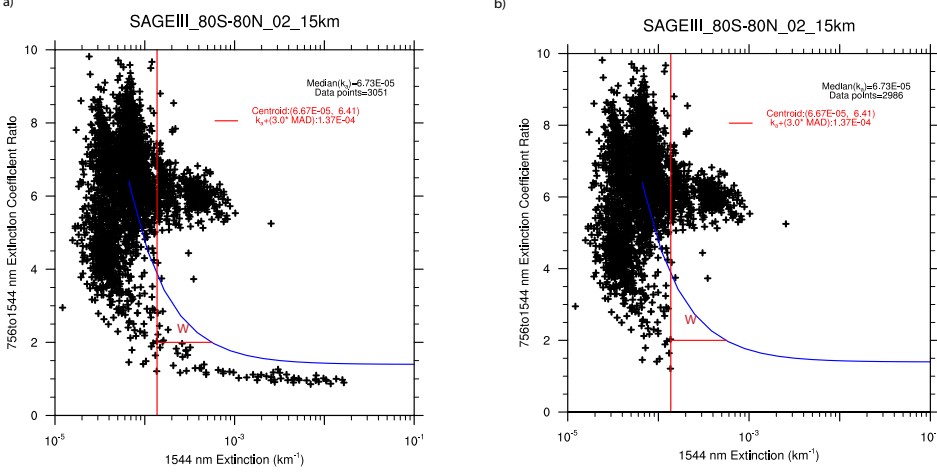

**Figure 3.** Scatter plots of 756 to 1544 nm extinction ratio as a function of 1544 nm extinction at 15 km altitude for February for the time period between 2017 and 2021, using TV13 Method. (a) before filtering and (b) after filtering.

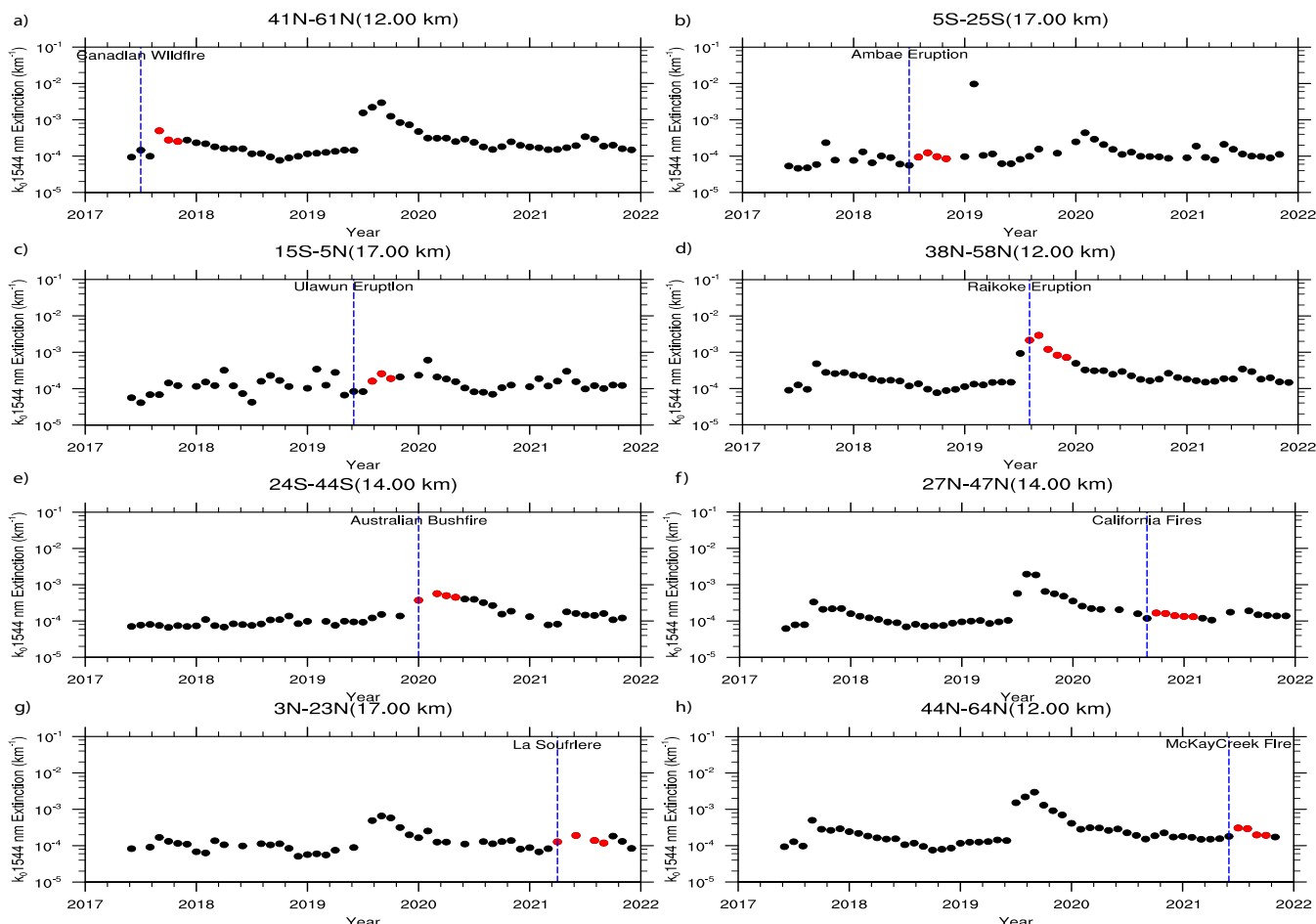

**Figure 4.** Time series of $k_0$ 1544 nm extinction for different latitude bands. Red symbols show the time line of enhanced aerosol extinction coefficient and the time it takes to get back to the background aerosol level following each event. The panels show events listed in Table 1 with (a) Canadian Wildfire, (b) Ambae eruption, (c) Ulawun Eruption, (d) Raikoke Eruption, (e) Australian Wildfire, (f) California Creek Fire, (g) La Soufriere and (h) McKay Creek Fire.

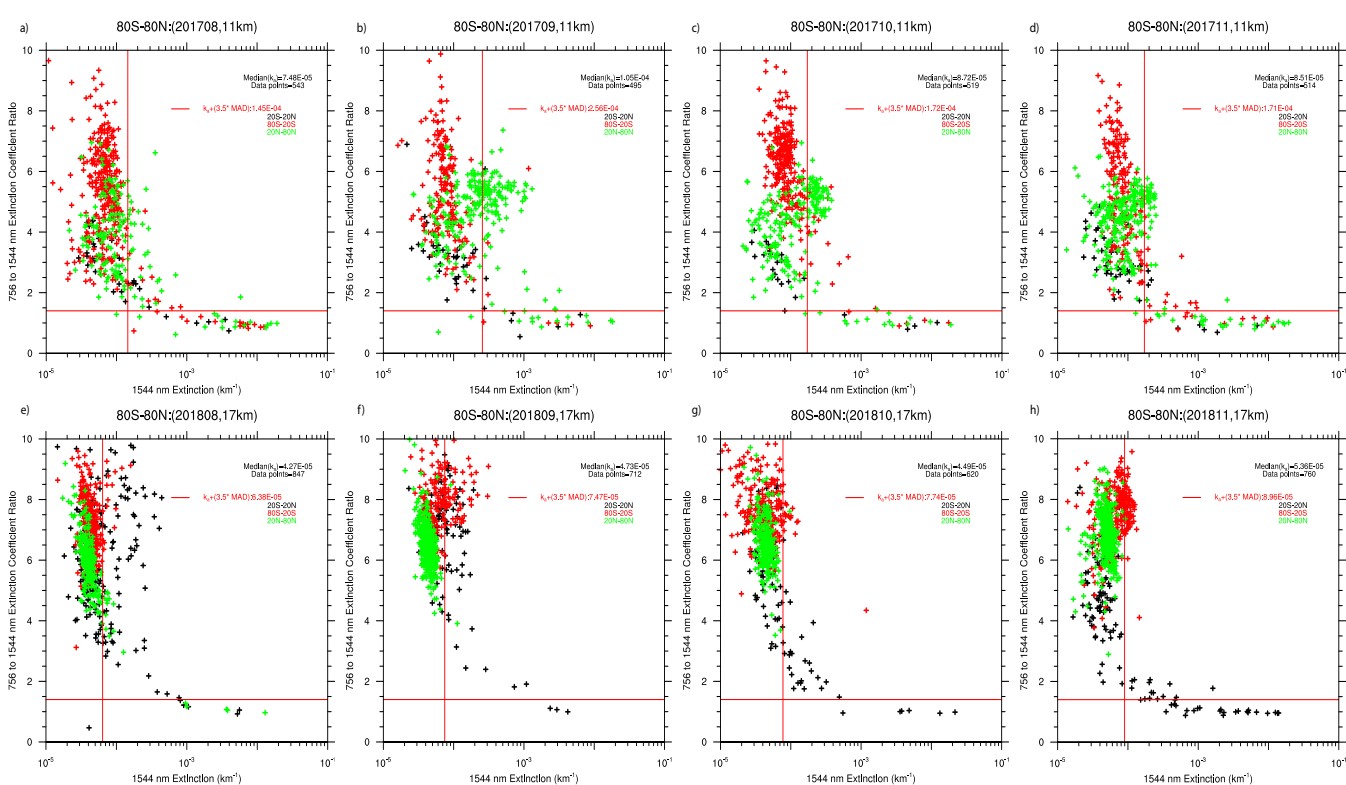

**Figure 5.** Scatter plots of 756 to 1544 nm extinction ratio versus 1544 nm extinction following Canadian Wildfire event in 201708 and Ambae eruption in 201807.



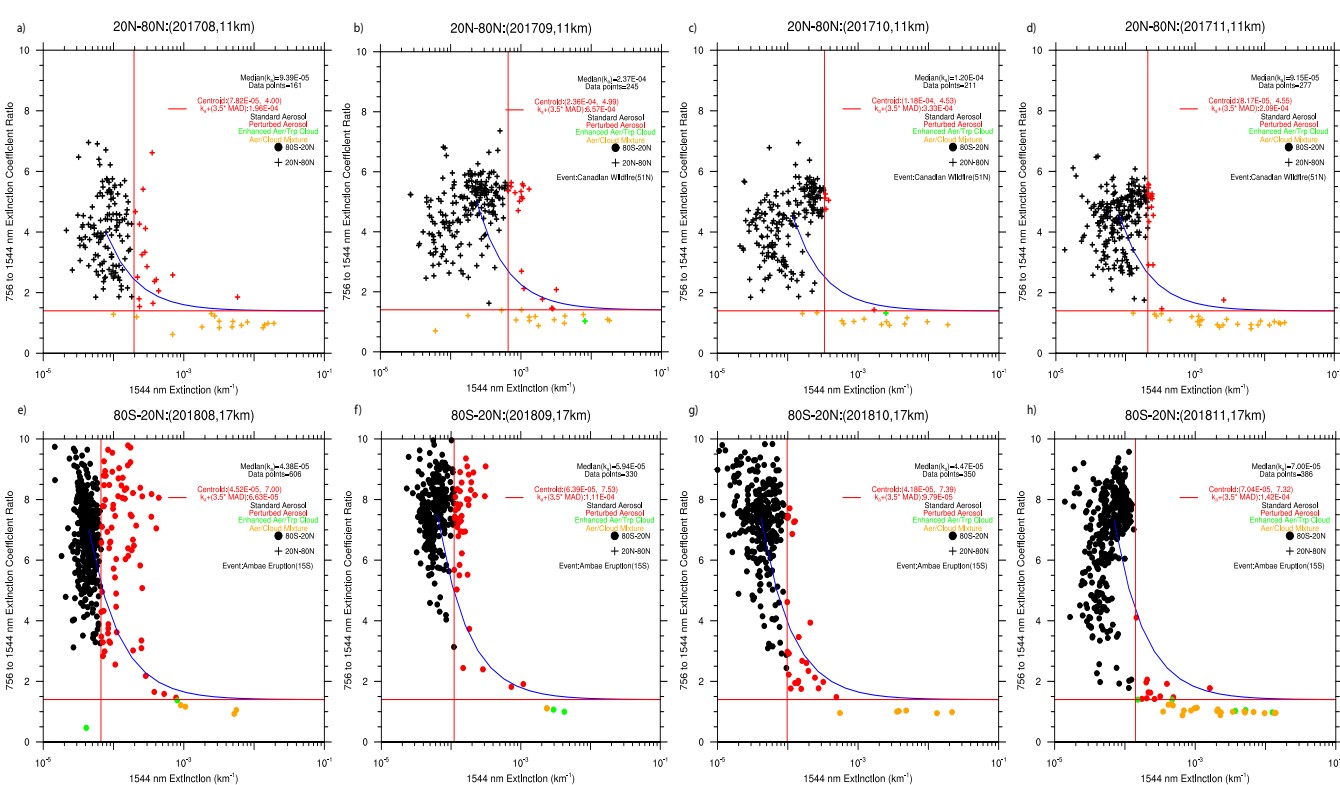

**Figure 6.** Scatter plots of 756 to 1544 nm extinction ratio versus 1544 nm extinction following Canadian Wildfire event in 201708 (a-d) and Ambae eruption in 201807 (e-h), after applying cloud/aerosol categorization.

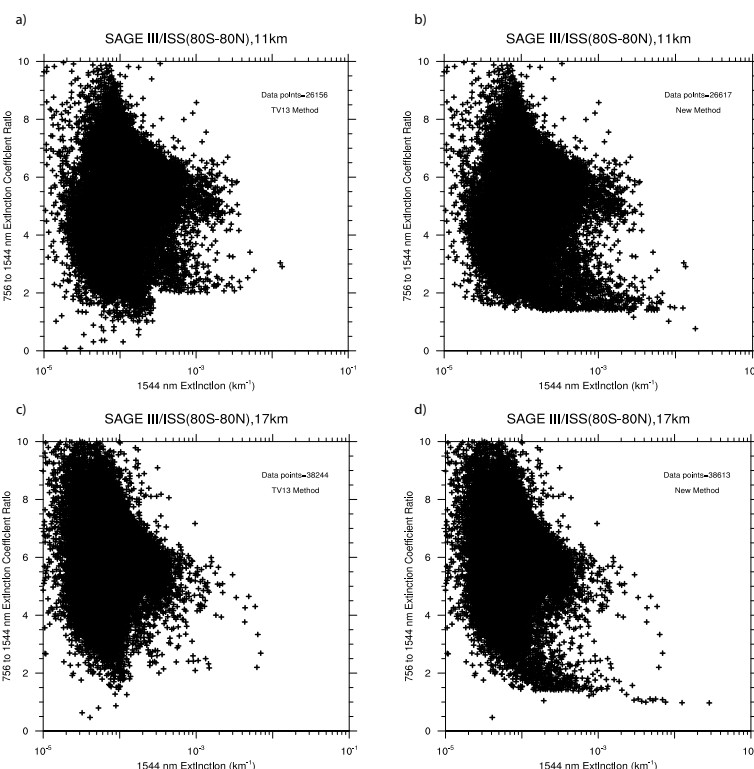

**Figure 7.** Scatter plots of 756 to 1544 nm extinction ratio as a function of 1544 nm extinction at 11 and 16km for the time period between 2017 and 2021, after filtering out possible aerosol/cloud mixture. (a,c) TV13 Method and (b,d) New Method.

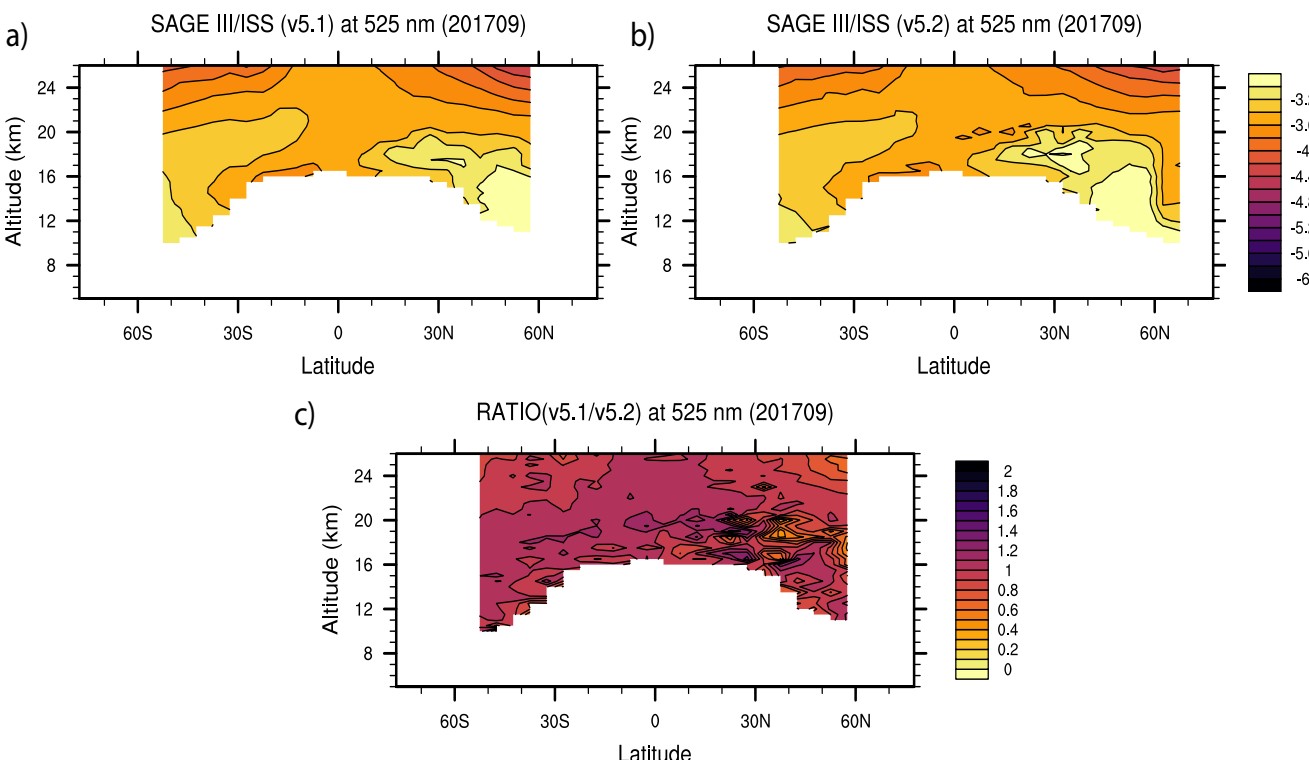

**Figure 8.** Zonally averaged SAGE III/ISS altitude versus latitude extinction coefficient plot for September 2017 following Canadian wildfire event. (a) version 5.1 (b) for version 5.2 and (c) ratio between version 5.2 and 5.1. Extinction coefficient values are shown in the log base to the 10.

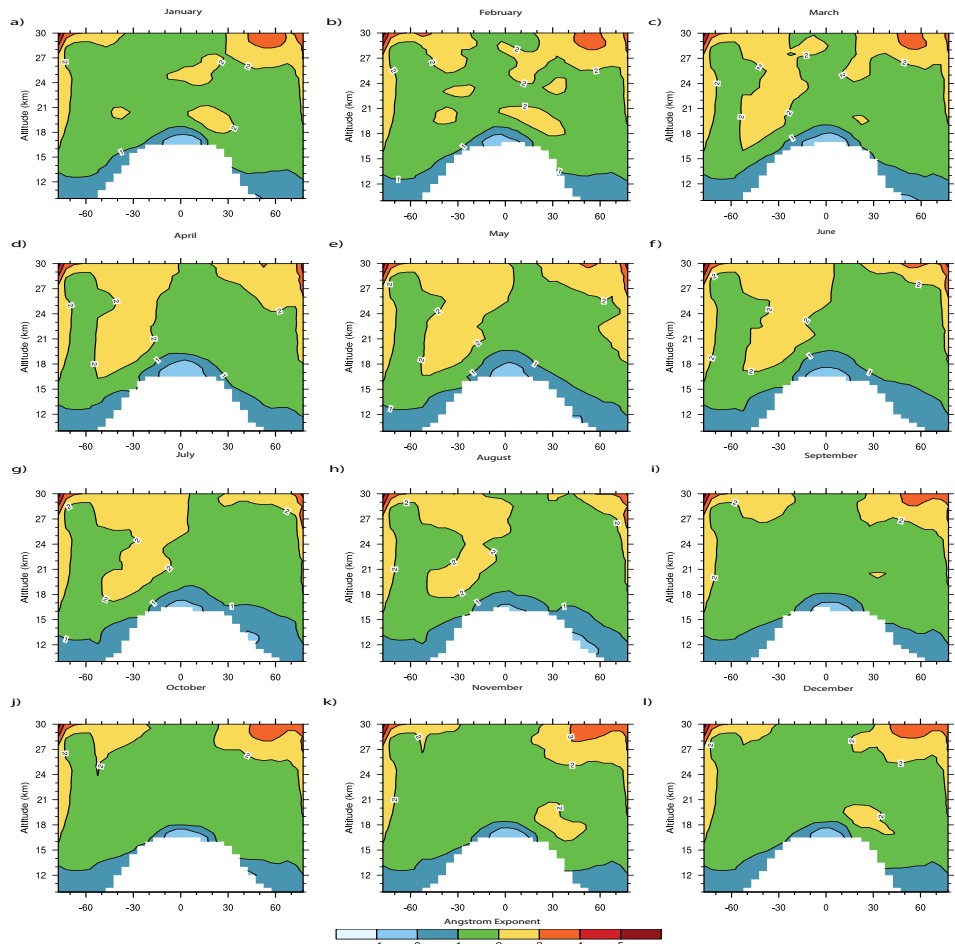

**Figure 9.** Altitude versus Latitude of Ångström exponent monthly climatology derived using OSIRIS 750 nm and SAGE II and SAGE III/ISS 525 nm extinction. Outliers are removed using 3x3 median smoothing. Please note that we apply linear interpolation to fill in missing data that are mostly applicable for the polar latitude (poleward of 55).



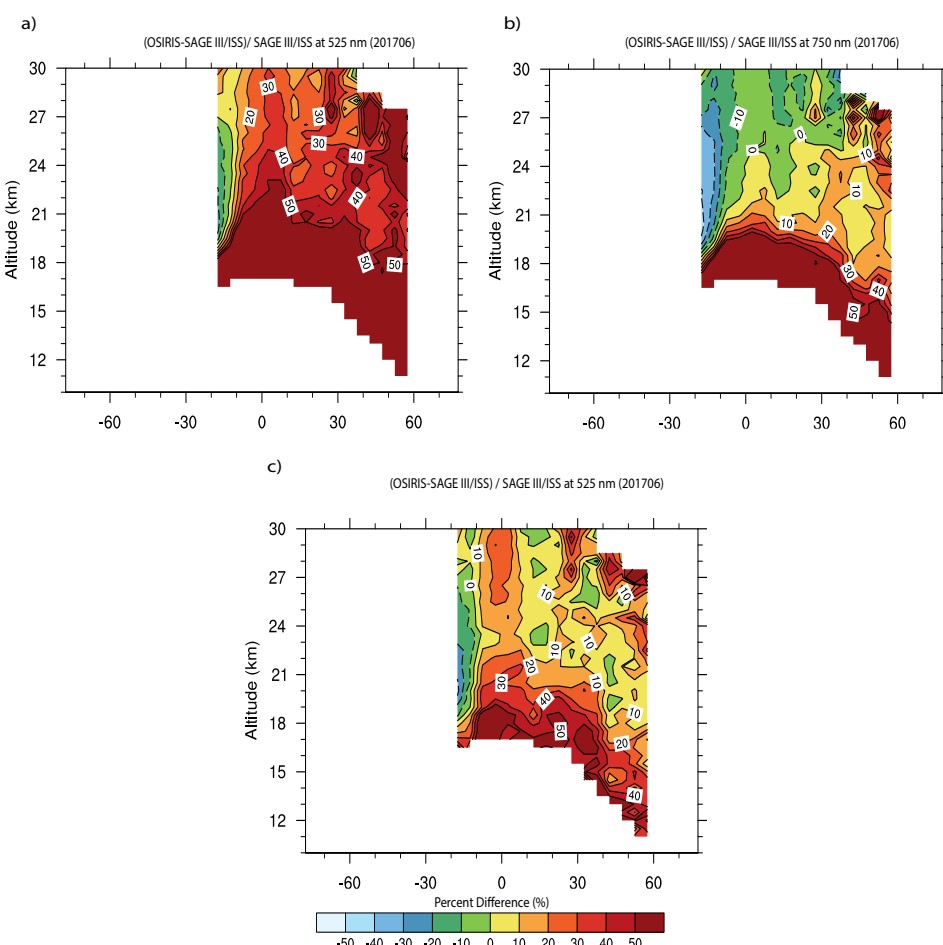

**Figure 10.** Percent difference between OSIRIS and SAGE III/ISS altitude versus latitude for June 2017 (a) at 525 nm , and (b) 750 nm. Ångström exponent of 2.33 is used to convert OSIRIS extinction to 525 nm in (a) while a monthly climatology of Ångström exponent from Figure 6 is used to convert OSIRIS extinction in (c).

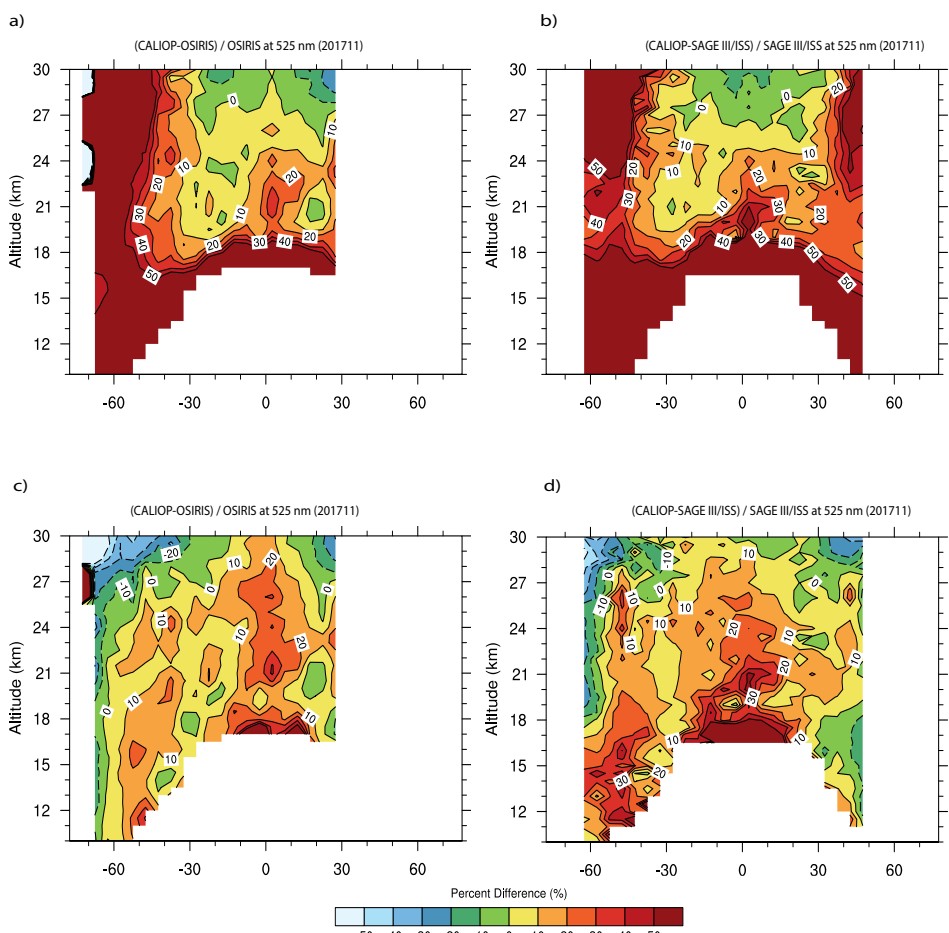

**Figure 11.** Percent difference between CALIOP, bias corrected OSIRIS, and cloud screened SAGE III/ISS extinction coefficients for November 2017. CALIOP data used in (a) and (b) are for 532 nm available in CALIOP stratospheric aerosol product, whereas CALIOP data in (c) and (d) are bias corrected using the scale factor (SF) showed in Figure 9a.





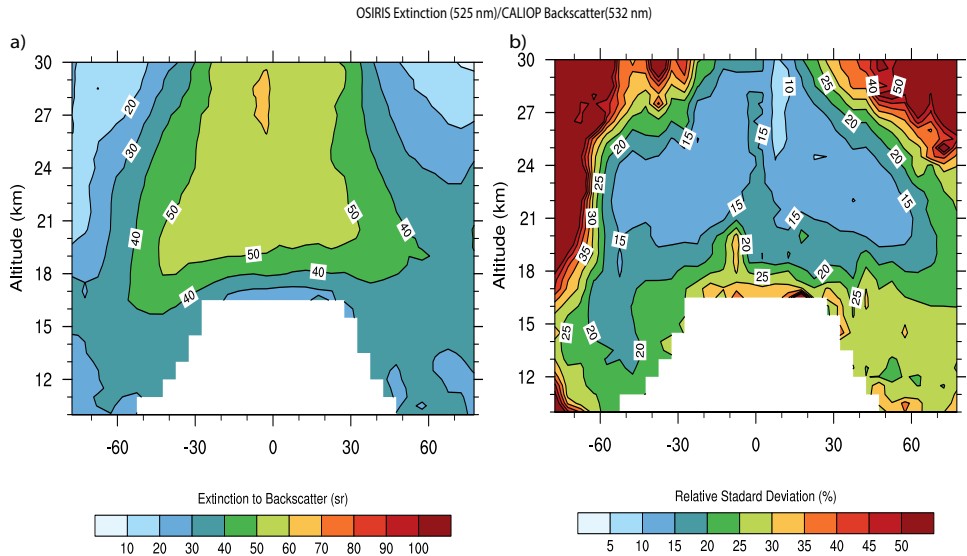

**Figure 12.** Altitude versus latitude dependence of 525 bias corrected OSIRIS extinction to 532 CALIOP backscatter ratio (SF) for the overlap period between 2006 and 2020.

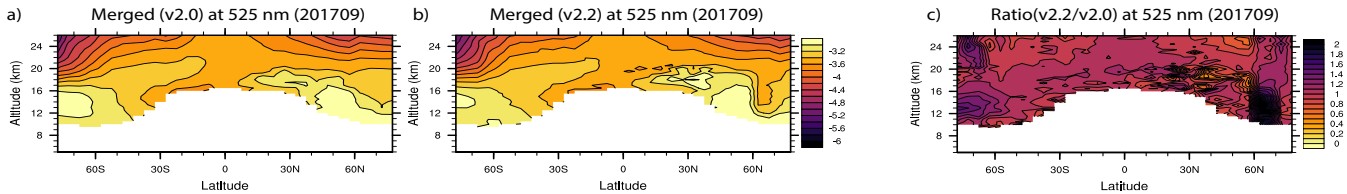

**Figure 13.** Altitude versus latitude dependence of 525 nm extinction for September 2017 and August 2018. (a) for GloSSAC version 2.0, (b) for GloSSAC version 2.2, and (c) ratio between version 2.2 and 2.0. Lower panels show same as in the upper panel but for August 2018.

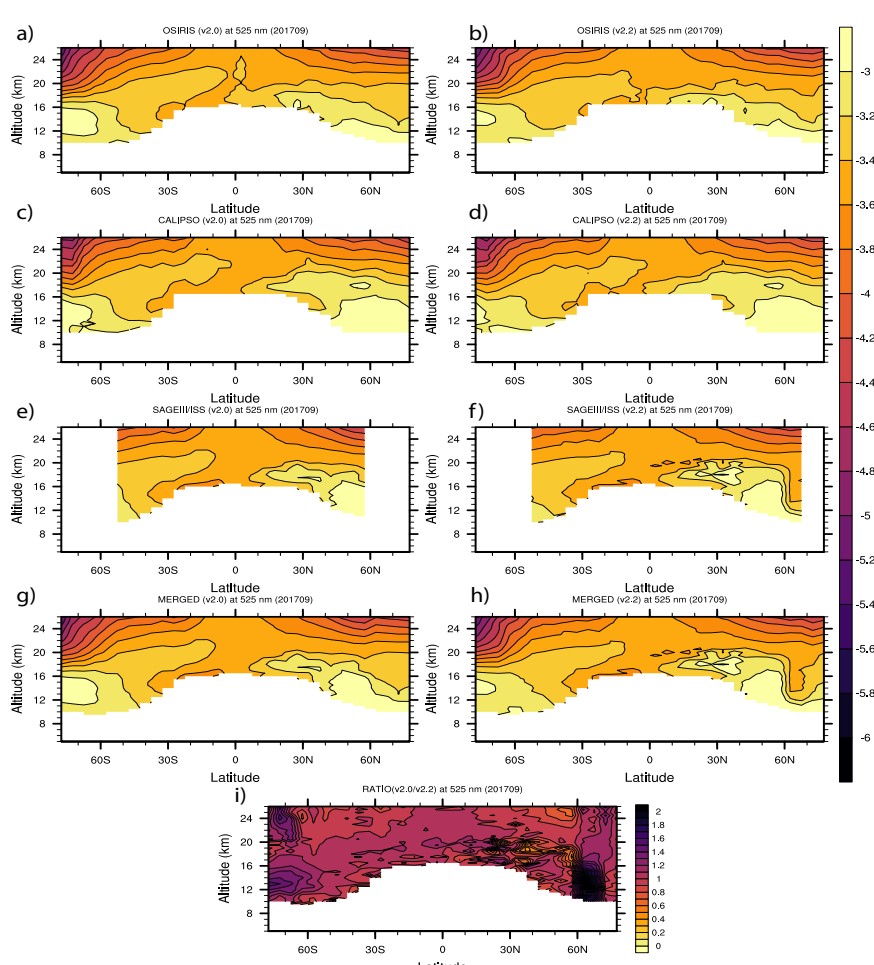

**Figure 14.** Altitude versus latitude dependence of 525 nm extinction for September 2017. (a,c,e,g) for OSIRIS, CALIPSO, SAGE III/ISS, and merged extinction for version 2.0 respectively whereas (b,d,f,h) are for version 2.2. (i) shows the ratio between merged version 2.2 and 2.0.



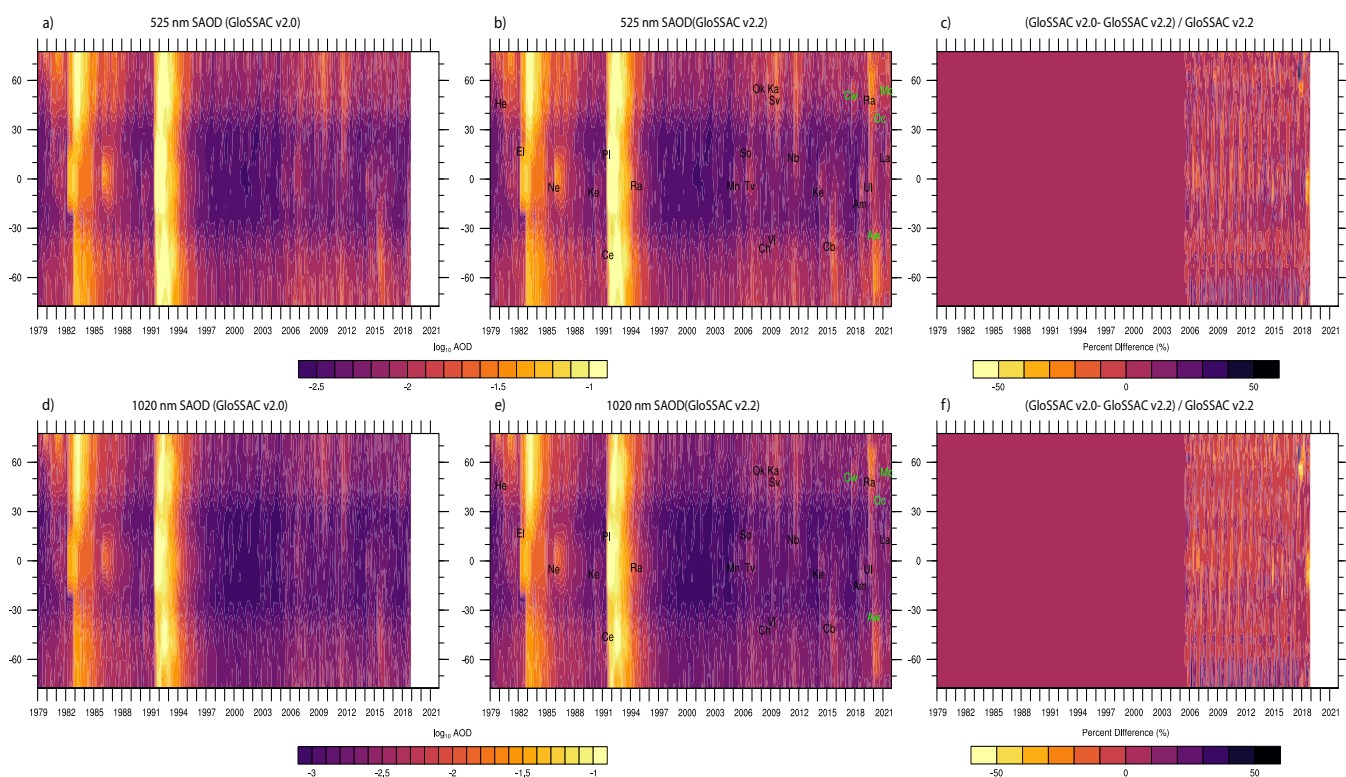

**Figure 15.** Latitude versus time dependence of SAOD for 525 and 1020 nm. (a,b,c) SAOD for 525 nm for GloSSAC version 2.0, version 2.2, and percent difference between (a) and (b). (d,e,f) Same as in (a,b,c) but for 1020 nm. (b, e) show major volcanic eruptions (black) and wild fire events (green) with abbreviated two letter code with their respective latitude and time of occurrence that are listed here. The event names shown in figures are: St. Helens (He), El Chichon (El), Nevado del Ruiz (Ne), Kelut (Ke), Pinatubo (Pi), Mt. Hudson (Ce),Rabaul (Ra), Manam (Mn), Soufriere Hills (So), Tavurvur (Tv), Chaiten (Ch), Okmok (Ok), Kasatochi (Ka), Sarychev (Sv), Nabro (Nb), Kelut (Ke), Calbuco (Cb), Canadian Wildfires (Cw), Ambae (Am), Ulawun (Ul), Australian Wildfire (Aw), California Creek Fire (Cc), La Soufriere (La), McKay Creek Fire (Mc).



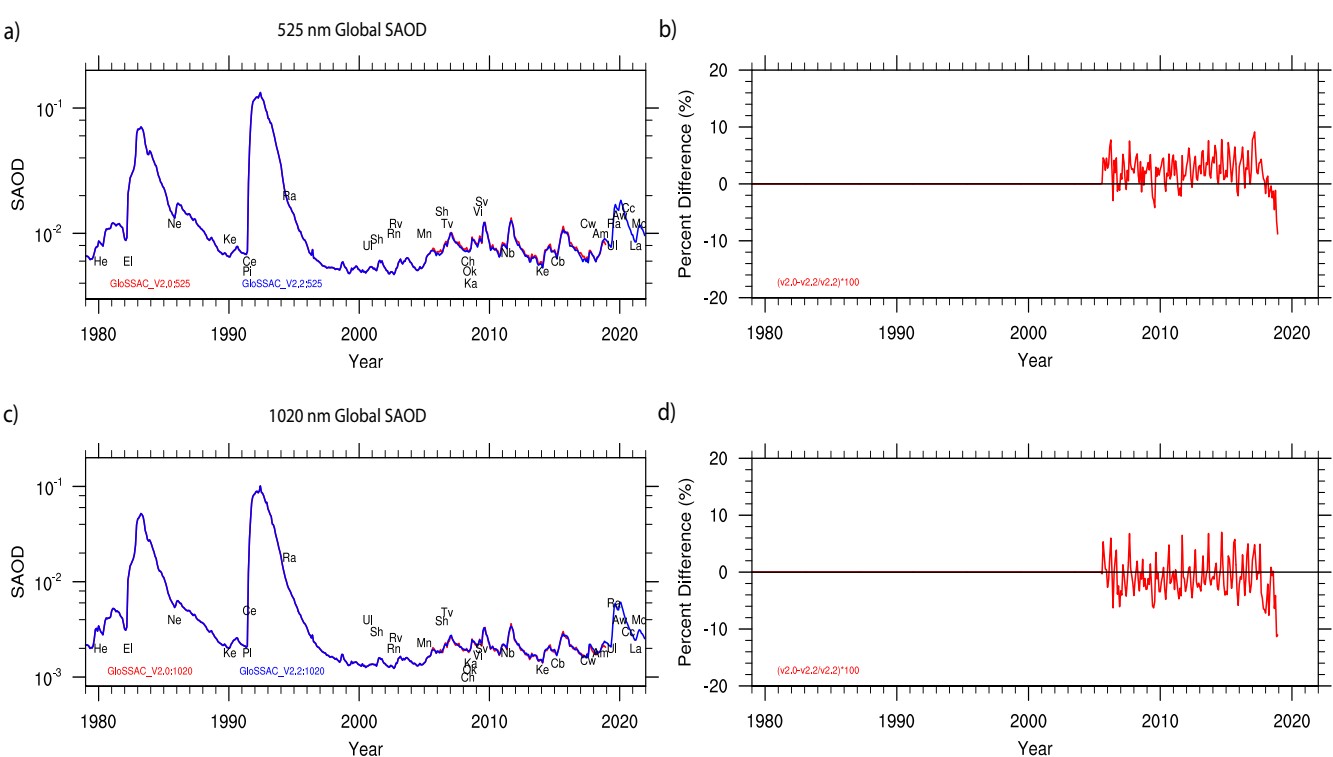

**Figure 16.** Time series of globally averaged SAOD for 525 and 1020 nm. Percent difference between version 2.0 and 2.2 are shown in (b,d) for 525 and 1020 nm.