# Peer review of "SAGE III/ISS aerosol/cloud categorization and its impact on GloSSAC"

_Atmospheric Measurement Techniques, 2022_

## Referee Comment (RC1)

**SAGE III/ISS aerosol/cloud categorization and its impact on GloSSAC**

Mahesh Kovilakam, Larry W. Thomason and Travis Knepp

This paper describes an evolution of the algorithm used to categorise SAGE III data, and is based very much on earlier papers by the same group. It should be a straightforward report-like paper describing how the new algorithm works and its impact on GloSSAC. Unfortunately the paper is woefully badly written, with some parts unintelligible, a missing description of a key model, inconsistencies between the figures and the text, and errors in English throughout. In particular, the definite article is omitted time after time – I have noted some of these cases below but gave up after p.5 l.120 as there are too many examples.

Did the co-authors actually read this manuscript before it was submitted? It is their responsibility to check the draft and ensure that it is clear, and grammatically correct. The revised manuscript needs to be checked carefully by a native English speaker. It is not the job of reviewers (or the journal) to do the authors' job for them.

Most of my comments below are minor, but a few are more substantial and there are so many of them as to require major revision of the manuscript before it can be accepted.

p.1 l.4-5 the stratosphere ……… the tropopause

p.1 l.10 locating the aerosol centroid

p.1 l.12 identifies

p.1. l.20 the solar occultation

p.3 l.57 'introduced' rather than 'occurred'

p.3 l.62 the Chappuis band

p.3 l.66 – 86 and fig.1. I do not understand the method used here. It seems from fig.1 that negative values above 30 km have been converted to positive values and retained? Is that correct? If so this change of sign must be very carefully justified because it biases any averages that are then made. Furthermore for the blue point at the tropopause on fig 1b four, not three, points have been removed from the set shown in fig 1a – so the diagram is not consistent with the text.

p.4 l.112 centroid and an

p.4 l.112 What is this empirical model supposed to be modelling? It turns out to be important (see below) yet you don't say what it is. A proper description is needed.

p.4 l.118-9. This sentence needs redrafting. For a start the factor is not 2, it is approximately 2, and that point needs to be made. But as written the sentence does not make sense – the bit after 'because' is not a valid reason. The reason is given in the next sentence – so some redrafting is needed here, especially as this sentence also (l.120) does not make sense. And in general you need 'the' before the number (e.g. 756) when it is being used as an adjective (e.g. 756 nm channel). This problem recurs throughout the manuscript, as mentioned above.

p.4 l.122 delete 'that'

p.5 l128 Delete the sentence 'Here,.......3a' as it is redundant

p.5. l.128 using the same method

p.5 l.128-130. This section is confusing. For a start, you've already said that figure 3 shows $k_0$ computed using the same method as TV13. So what is the point of the next sentence, which just repeats the same thing? And as you haven't said anything about the empirical model previously, other than to mention its existence on l. 112, the sentence 'Following TV13 method, we use an offset of 0.4 for the empirical model ratios.......' is unintelligible. This model needs a proper description (with equations).

p.5 l.131. This appears to be saying that points between the red and blue lines are aerosol/cloud mixtures. So what is the point of the next two sentences? Are they not just repeating what you've just said (other than the last part of the second sentence? This part of the paper is **really** badly written.

p.5 l.132 Assuming that the blue line in fig 3 is the empirical model, what significance should be attached to the crossing of the blue and red lines, and why are there so many points to the right of both of them above an extinction ratio of 5?

p.5 l.136 relatively

p.5 l.139. What I think you are saying is that the long tail below an extinction ratio of 2, extending to higher extinction, is actually aerosol. You give no proof at all of this statement – have you examined some individual cases to validate that they are not cloud-affected? If not, how do you know they are aerosol?

p.5 l.143-4 Please point out to the reader where exactly these clusters are on fig 3, as it's not clear to me. Also, 'there appear' not 'there appears'

p.6 l.163 statistics are

p.7 l.193 points that fall

p.7 l.196 Surely you mean the ratio rather than the extinction coefficient?

p.7 l.198 there is more than one

p.7 l.211 The first sentence here is redundant – it just repeats what's already been said.

p.7 l.213 These are the same events as you've already discussed. Refer to the previous discussion instead of introducing them as completely new events.

p.7 l.216-8 The horizontal red line also appears in fig.5 but is not described either in the text or the figure caption. Either this description should be moved to the discussion of fig.5 or the horizontal line is redundant in that figure and should be removed. In any case, what does it mean? Are you trying to define a threshold below which the ratio represents a mixture of cloud and aerosol? If so you need to say so explicitly (this issue of course follows on from the lack of explanation of the scattering model on p.4).

p.7 l.219 'Where' rather than 'Since', also delete 'then' in l.220

p.8 l.242. Here you say that the green symbols in fig 6 are categorised as 'enhanced aerosol/tropopause cloud'. In the previous paragraph it is clearly stated (l.221 on) that this category applies to points below the red horizontal line (lower two quadrants). My understanding of this

paragraph is that the time series of $k_0$ after the event was used to define a perturbed period when different rules would apply in this part of the diagram. Yet on l.242 we are told that **all** the data measured in these periods are categorised as 'enhanced aerosol/tropopause cloud', because all the 20N – 80N points in these panels are green. Surely you have just used green as a geographic label in fig.6?

p.8 l.245 What empirical model? I can't see anything that looks like a model in fig.6. Indeed I don't understand what the text from l.245-249 is saying. There is a cloud cluster with a centroid extinction of $10^{-1}$ $km^{-1}$? That's the right-hand axis of the panels in fig 6, where there are no data points.

p.9 l.265  the Coupled

p.9 l.268 delete 'While other……..era'

p.9 l.271-275. The sentence 'It is known……UTLS region' just repeats earlier text and is redundant – remove it.

p.10 l. 296 The sentence 'However, by using the timeframe shown in the monthly time series of $k_0$ in Figure 4 could alleviate the bias to some extent' is either redundant or needs redrafting – how does the timeframe 'alleviate the bias'?

p.10 l.297 – here you show extinction coefficients at 525 nm, the wavelength where there is a negative bias (l.122),  and not the wavelength used for the results presented so far. What is the point of this figure if it doesn't relate to anything shown so far in the paper? The reason, it seems, is that GLOSSAC uses 525 and 1020 nm. In that case this section must say so, and explain how the SAGE-III data at 525 nm were measured (or calculated). As it stands, you have spent the first part of the paper discussing the analysis of data at 756 and 1544 nm which is not used in the rest of the paper. How does this and subsequent sections relate to the earlier sections?

p.10 l.300 There is very little difference in the data shown on 8a and 8b, as indeed 8c shows. The most striking difference is that the data in 8b extend to higher latitude. This is completely different to what the text describing this figure says.

p.10 l.308 extended through

p.12 l.357 delete 'that are'

p.12 l.365. If 'reasonable agreement' is defined as <20% then almost all of fig.10a falls outside this definition. The sentence applies only to 10b.

p.12 l.367 remove 'in comparison with Figure 10b' (repeats)

p.12 l.369. No, the comparison with OSIRIS does not show 'reasonable agreement' in 10a. This whole paragraph is confused, and not consistent with the figure. You must base your argument on the facts you present, not on wishful thinking.

p.12 l.380. Recently, it has been shown…….. eruptions were manifest in the SAGEII…..

p.12 l.381 'show a decrease' (remove 'that')

p.12 l.380-388 What do you mean by 'inferring aerosol size .. for the post-SAGEII period .. deficient'? From lines 330-331, the data from 2005-17 came from OSIRIS (750 nm) and CALIOP (532 nm). Do you mean that these wavelengths are too close together to provide a reliable size estimate? Converting them to 525/1050 with some arbitrary Angstrom coefficient is a smokescreen. This

paragraph doesn't come to any useful conclusion – are we to take it that the size estimates for 2005-17 are no good?

p.13 l.399. The data in fig 11 cover -70-+27° latitude and are unlikely to be much affected by PyroCb at 50-60°N.

p.13 l.406 in the same way

p.14 l.427-430. First of all, there is another spurious 'that' in this sentence, between v2.2 and apparently. Secondly, the statement made in this sentence is not evident to me, looking at the figures. Please point out more clearly where on the figures you are referring to – not the difference >60° it seems, as you go on to discuss that separately. And thirdly, how can you have 'clearly' and 'apparently' in the same sentence? Which is it?

p.14 l.431. This time 'that' has been omitted (changes that occurred).

p.14 l.431 Fig 14 can be removed as this is the only (fleeting) reference to it in the paper.

p.15 l.455 are otherwise

p.15 l.475 what does 'relatively the same' mean? How is this different to 'the same'?

p.16 l.485. This is the first mention of CLAES so where does the 'presumably underestimating aerosol extinction' comment come from? And why is it relevant?

Fig 1 caption l.1 remove 'how'

Figs 2, 3, 5, 6, 7 increase size of labels on the plots: they should be readable on a laptop screen without zooming right in

Fig 3 caption – explain what the red and blue lines are

Fig 5 caption – what is the horizontal red line? Explain which panels show data from which period, and point out the difference in height for the two events. Try to help the reader understand your paper!

Fig 6 caption – point out the differences between this and fig.5 e.g. different latitude bands and algorithms. Explain clearly what all the lines and points mean.

Fig 8 caption: log to base 10

Fig 9 increase size of axis labels and contour labels. In caption, state the period of the climatology

Fig 11 caption says that 'CALIOP data in (c)and (d) are bias corrected using the scale factor (SF) showed in Figure 9a.' But fig 9 shows Angstrom coefficients not scale factors, and according to the text (l.401-406) the bias correction comes from another paper by this group. You must ensure consistency between text and figures.

---

## Author Comment (AC1)

**Response to Reviewer #1**

We thank the reviewer for helpful comments. Our responses to the reviewer's specific comments are listed below. The reviewer's concerns are in bold italicized font and our responses are in regular font. The page numbers and line numbers given in our responses below are in reference to the revised version of the manuscript.

***This paper describes an evolution of the algorithm used to categorise SAGE III data, and is based very much on earlier papers by the same group. It should be a straightforward report-like paper describing how the new algorithm works and its impact on GloSSAC. Unfortunately the paper is woefully badly written, with some parts unintelligible, a missing description of a key model, inconsistencies between the figures and the text, and errors in English throughout. In particular, the definite article is omitted time after time ? I have noted some of these cases below but gave up after p.5 l.120 as there are too many examples. Did the co-authors actually read this manuscript before it was submitted? It is their responsibility to check the draft and ensure that it is clear, and grammatically correct. The revised manuscript needs to be checked carefully by a native English speaker. It is not the job of reviewers (or the journal) to do the authors? job for them. Most of my comments below are minor, but a few are more substantial and there are so many of them as to require major revision of the manuscript before it can be accepted.***

**Specific comments**

***p.1 l.4-5 the stratosphere ??? the tropopause***

Done. Thanks.

***p.1 l.10 locating the aerosol centroid***

Done. Thanks.

***p.1 l.12 identifies***

Done. Thanks.

***p.1. l.20 the solar occultation***

Done. Thanks.

***p.3 l.62 the Chappuis band***

Done.

*p.3 l.66 ? 86 and fig.1. I do not understand the method used here. It seems from fig.1 that negative values above 30 km have been converted to positive values and retained? Is that correct? If so this change of sign must be very carefully justified because it biases any averages that are then made. Furthermore for the blue point at the tropopause on fig 1b four, not three, points have been removed from the set shown in fig 1a ? so the diagram is not consistent with the text.*

The extinction profile shown in Figure 1 represents absolute extinction coefficient. All extinction values are plotted as absolute values and negative extinction values are color coded using red (blue) filled circles with $> 50\%$ ($< 50\%$) error, whereas orange symbols represent positive extinction with $> 50\%$ error. This information is now added to the figure caption as well as in the text. Yes, thanks for pointing out the discrepancy in the figure. There was a small bug in the code, which is now corrected and revised. The figure is now revised and shows only three points have been removed. We do not remove any negative extinction coefficients above 25 km as they mostly occur due to noise and errors in the removal of ozone and molecular scattering and thus are retained in the data as shown in Figure 1.

*p.4 l.112 centroid and an*

Done. The description of empirical model is now moved to supplementary section (S1).

*What is this empirical model supposed to be modelling? It turns out to be important (see below) yet you don?t say what it is. A proper description is needed.*

The empirical model can be fitted to the observed data as shown in Section S1 of the supplementary. We rewrote this section and moved this section to supplementary.

*p.4 l.118-9. This sentence needs redrafting. For a start the factor is not 2, it is approximately 2, and that point needs to be made. But as written the sentence does not make sense ? the bit after ?because? is not a valid reason. The reason is given in the next sentence ? so some redrafting is needed here, especially as this sentence also (l.120) does not make sense. And in general you need ?the? before the number (e.g. 756) when it is being used as an adjective (e.g. 756 nm channel). This problem recurs throughout the manuscript, as mentioned above.*

We have revised this section and now use 525:1020 nm extinction ratio, same as TV13. The method is explained in detail in section 3.2 .

*p.4 l.122 delete 'that'*

Done. Thanks.

*p.5 l128 Delete the sentence 'Here, ...... 3a' as it is redundant*

This section is rewritten and moved to Supplementary (section S1).

**_p.5. l.128 using the same method_**

This section is rewritten and moved to Supplementary (section S1).

**_p.5 l.128-130. This section is confusing. For a start, you've already said that figure 3 shows k0 computed using the same method as TV13. So what is the point of the next sentence, which just repeats the same thing? And as you haven?t said anything about the empirical model previously, other than to mention its existence on l. 112, the sentence ?Following TV13 method, we use an offset of 0.4 for the empirical model ratios??.? is unintelligible. This model needs a proper description (with equations)._**

The entire section has been rewritten and we now use flowchart to discuss the method involved. The description of empirical model is now moved to supplementary section (S1) with equations.

**_p.5 l.131. This appears to be saying that points between the red and blue lines are aerosol/cloud mixtures. So what is the point of the next two sentences? Are they not just repeating what you?ve just said (other than the last part of the second sentence? This part of the paper is really badly written._**

The entire section (now section 3.2) has been rewritten to make things clearer with the help of flowchart (section 3.2).

**_p.5 l.132 Assuming that the blue line in fig 3 is the empirical model, what significance should be attached to the crossing of the blue and red lines, and why are there so many points to the right of both of them above an extinction ratio of 5?_**

The vertical red line shows the separation of background and perturbed aerosols. So, data points that fall to the right of red line with ratio greater than 1.4 are considered to be perturbed aerosols in this case. We made it clearer in the revised version. Please note that this section has been moved to supplementary (S1, lines 21-25).

**_p.5 l.136 relatively_**

This sentence has been removed.

**_p.5 l.139. What I think you are saying is that the long tail below an extinction ratio of 2, extending to higher extinction, is actually aerosol. You give no proof at all of this statement ? have you examined some individual cases to validate that they are not cloud-affected? If not, how do you know they are aerosol?_**

This sentence has been removed from here to avoid any confusion. The entire section is now rewritten to make things clearer. It is now discussed in section 3.3

***p.5 l.143-4 Please point out to the reader where exactly these clusters are on fig 3, as it?s not clear to me. Also, ?there appear? not ?there appears?***

This is now discussed in the supplementary section (S1, line 6-7 ).

***p.6 l.163 statistics are***

Done. Thanks.

***p.7 l.193 points that fall***

This sentence has been rewritten.

***p.7 l.196 Surely you mean the ratio rather than the extinction coefficient?***

Changed it to extinction ratios.

***p.7 l.198 there is more than one***

Moved to supplementary section S2 (lines 31-35).

***p.7 l.211 The first sentence here is redundant ? it just repeats what?s already been said.***

This has been removed.

***p.7 l.213 These are the same events as you?ve already discussed. Refer to the previous discussion instead of introducing them as completely new events.***

This section has been rewritten.

***p.7 l.216-8 The horizontal red line also appears in fig.5 but is not described either in the text or the figure caption. Either this description should be moved to the discussion of fig.5 or the horizontal line is redundant in that figure and should be removed. In any case, what does it mean? Are you trying to define a threshold below which the ratio represents a mixture of cloud and aerosol? If so you need to say so explicitly (this issue of course follows on from the lack of explanation of the scattering model on p.4).***

This figure and this section have been revised.

***p.7 l.219 ?Where? rather than ?Since?, also delete ?then? in l.220***

This has been removed and the section is rewritten using a flowchart. .

*p.8 l.242. Here you say that the green symbols in fig 6 are categorised as ?enhanced aerosol/tropopause cloud?. In the previous paragraph it is clearly stated (l.221 on) that this category applies to points below the red horizontal line (lower two quadrants). My understanding of this paragraph is that the time series of k0 after the event was used to define a perturbed period when different rules would apply in this part of the diagram. Yet on l.242 we are told that all the data measured in these periods are categorised as ?enhanced aerosol/tropopause cloud?, because all the 20N ? 80N points in these panels are green. Surely you have just used green as a geographic label in fig.6?*

This section has been rewritten (lines 190-196).

*p.8 l.245 What empirical model? I can?t see anything that looks like a model in fig.6. Indeed I don?t understand what the text from l.245-249 is saying. There is a cloud cluster with a centroid extinction of 10-1 km-1? That?s the right-hand axis of the panels in fig 6, where there are no data points.*

The empirical model section has been moved to supplementary (section S2).

*p.9 l.265 the Coupled*

Corrected (line 281).

*p.9 l.268 delete ?While other??..era?*

Removed. Thanks.

*p.9 l.271-275. The sentence ?It is known??UTLS region? just repeats earlier text and is redundant ? remove it.*

Removed. Thanks.

*p.10 l. 296 The sentence ?However, by using the timeframe shown in the monthly time series of k0 in Figure 4 could alleviate the bias to some extent? is either redundant or needs redrafting ? how does the timeframe ?alleviate the bias??*

Removed. Thanks.

*p.10 l.297 ? here you show extinction coefficients at 525 nm, the wavelength where there is a negative bias (l.122), and not the wavelength used for the results presented so far. What is the point of this figure if it doesn?t relate to anything shown so far in the paper? The reason, it seems, is that GLOSSAC uses 525 and 1020 nm.*

*In that case this section must say so, and explain how the SAGE-III data at 525 nm were measured (or calculated). As it stands, you have spent the first part of the paper discussing the analysis of data at 756 and 1544 nm which is not used in the rest of the paper. How does this and subsequent sections relate to the earlier sections?*

This has been revised and included explanation on why 525 nm is chosen for GloSSAC (lines 427-434).

*p.10 l.300 There is very little difference in the data shown on 8a and 8b, as indeed 8c shows. The most striking difference is that the data in 8b extend to higher latitude. This is completely different to what the text describing this figure says.*

While the differences are generally small between the two versions, in the lower stratosphere for the latitudes between 37.5 and 57.5 degrees N, show lower ratios between 0.40 and 0.6 that suggests clear enhancement of extinction in version 5.2. This section has been rewritten (lines 435-440).

*p.10 l.308 extended through*

Done.

*p.12 l.357 delete ?that are?*

Deleted and rewritten (lines 356-360).

*p.12 l.365. If ?reasonable agreement? is defined as <20% then almost all of fig.10a falls outside this definition. The sentence applies only to 10b.*

This has been rewritten to make it clearer (lines 347-350).

*p.12 l.367 remove ?in comparison with Figure 10b? (repeats)*

Done (line 347).

*p.12 l.369. No, the comparison with OSIRIS does not show ?reasonable agreement? in 10a. This whole paragraph is confused, and not consistent with the figure. You must base your argument on the facts you present, not on wishful thinking.*

This has been rewritten.

*p.12 l.380. Recently, it has been shown??.. eruptions were manifest in the SAGEII?..*

It now reads as "Recently, it has been shown that many small to moderate eruptions were manifest during SAGE II and III/ISS data " (line 377)

***p.12 l.381 ?show a decrease? (remove ?that?)***

Done.

***p.12 l.380-388 What do you mean by ?inferring aerosol size .. for the post-SAGEII period .. deficient?? From lines 330-331, the data from 2005-17 came from OSIRIS (750 nm) and CALIOP (532 nm). Do you mean that these wavelengths are too close together to provide a reliable size estimate? Converting them to 525/1050 with some arbitrary Angstrom coefficient is a smokescreen. This paragraph doesn?t come to any useful conclusion ? are we to take it that the size estimates for 2005-17 are no good?***

We are using measurements to estimate Angstrom exponent climatology as showed in Figure 12. As mentioned in lines 372-376, the angstrom exponent conversion technique does not work effectively when the stratosphere is in the perturbed state. The challenges in converting extinction from one wavelength to another, particularly with a single wavelength measurement is deficient in inferring aerosols following perturbed events. For OSIRIS, we only have extinction at 750 nm and for CALIPSO it is 532 nm backscatter. So, both these measurements provide extinction/backscatter coefficient at a single wavelength that is not adequate to infer size information. So, the challenge we have here is to account for evolving size changes following any perturbed event and fill in the gaps effectively when multiwavelength measurements are not available. We do this by using the conversion technique mentioned in the manuscript and also in Kovilakam et al. (2020). So, we are being transparent here by stating that there is deficiency in this method, particularly for the period between 2005 and 2017, where we only have single wavelength measurement available from OSIRIS/CALIPSO. Therefore, with the data we have, this is the maximum we could do and it is not a smokescreen.

***p.13 l.399. The data in fig 11 cover -70-+27 $^0$ latitude and are unlikely to be much affected by PyroCb at 50-60$^0$N.***

It is now rewritten as "While we can attribute some of these differences in the northern higher latitudes to PyroCb event associated with Canadian wildfire" (lines 397-398).

***p.13 l.406 in the same way***

Done.

***p.14 l.427-430. First of all, there is another spurious ?that? in this sentence, between v2.2 and apparently. Secondly, the statement made in this sentence is not evident to me, looking at the figures. Please point out more clearly where on the figures you are referring to ? not the difference ¿60$^0$ it seems, as you go on to discuss that separately. And thirdly, how can you have ?clearly? and ?apparently? in the same sentence? Which is it?***

Corrected the sentence and rewritten the paragraph (lines 449-455).

**_p.14 l.431. This time ?that? has been omitted (changes that occurred)._**

Done. Thanks.

**_p.14 l.431 Fig 14 can be removed as this is the only (fleeting) reference to it in the paper._**

We kept this figure and removed Figure 13 as we would like to show version changes in the individual measurements.

**_p.15 l.455 are otherwise_**

Done.

**_p.15 l.475 what does ?relatively the same? mean? How is this different to ?the same??_**

Removed "relatively".

**_p.16 l.485. This is the first mention of CLAES so where does the ?presumably underestimating aerosol extinction? comment come from? And why is it relevant?_**

We have now revised this part and it now reads as: "We are currently revisiting this method to identify smoke events for SAGE II. In light of the new insights in the development of this new technique, we will likely revisit cloud detection used for the SAGE II in the production of the GloSSAC data set."

**_Fig 1 caption l.1 remove ?how?_**

Done.

**_Figs 2, 3, 5, 6, 7 increase size of labels on the plots: they should be readable on a laptop screen without zooming right in_**

Done.

**_Fig 3 caption ? explain what the red and blue lines are_**

Done. This section has been moved to supplementary.

**_Fig 5 caption ? what is the horizontal red line? Explain which panels show data from which period, and point out the difference in height for the two events. Try to_**

*help the reader understand your paper!*

Removed those lines from figures and revised the caption. Thanks.

*Fig 6 caption ? point out the differences between this and fig.5 e.g. different latitude bands and algorithms. Explain clearly what all the lines and points mean.*

This section has been rewritten using a flowchart and figure. Thanks.

*Fig 8 caption: log to base 10*

Done.

*Fig 9 increase size of axis labels and contour labels. In caption, state the period of the climatology*

Done. Thanks.

*Fig 11 caption says that ?CALIOP data in (c)and (d) are bias corrected using the scale factor (SF) showed in Figure 9a.? But fig 9 shows Angstrom coefficients not scale factors, and according to the text (l.401-406) the bias correction comes from another paper by this group. You must ensure consistency between text and figures.*

Corrected. Thanks.

**References**

Kovilakam, M., Thomason, L. W., Ernest, N., Rieger, L., Bourassa, A., and Millán, L.: The Global Space-based Stratospheric Aerosol Climatology (version 2.0): 1979–2018, Earth System Science Data, 12, 2607–2634, https://doi.org/10.5194/essd-12-2607-2020, 2020.

---

## Author Comment (AC2)

**Response to Reviewer #2**

We thank the reviewer for helpful comments. Our responses to the reviewer's specific comments are listed below. The reviewer's concerns are in bold italicized font and our responses are in regular font. The page numbers and line numbers given in our responses below are in reference to the revised version of the manuscript.

***The manuscript ?SAGE III/ISS aerosol/cloud categorization and its impact on GloSSAC? describes a new method of cloud screening for SAGE III/ISS spectral aerosol extinction observations and how this changes its time series, how the new data set compares with correlative observations (OSIRIS and CALIOP) and, finally, how this modification impacts on the GloSSAC merged time series ? one crucial component of which is the SAGE III/ISS data set. The topic of the manuscript clearly is of interest for the AMT readership. Nevertheless, I agree with the Anonymous Referee #1 that this manuscript is not clear, not well organised and this requires a very systematic work of synthesis, re-working of the text and clarification in many places before it can be considered for publication on AMT. During my review, I struggled to follow the flow of ideas that drive the manuscript and so I can only recommend a deep re-structuring and re-writing of the paper before I can actually review it. To help out with this, I add a number of suggestions which are not to be considered a full review but just examples of some modifications that are need throughout the text. A more general comment is that, in my opinion, there are in general too many figures and the overall paper is too wordy. I would suggest to distil the most informative information in terms of both methodology and results and transfer part of the reduntant information in a Supplementary Material section of the manuscript. Another general major comment is that, while the methodological aspects are described in very detailed way (in fact, overlenghty ? I suggest distilling most of this in a flow-chart figure or something similar), the results are discussed very superficially, with a large collection of figures with scarce or even no insights and discussions. Results must be analysed more in depth, to identify the differences of the new method SAGEIII versus the old one and, especially, the impact on GloSSAC and the differences of v2.0 versus v2.2. My best regards***

**Specific comments**

***L27: Why not citing more recent papers on stratospheric-aerosol-mediated impacts on the radiative balance? (Raikoke 2019: https://acp.copernicus.org/articles/21/535/2021/ Australian fires 2020: https://acp.copernicus.org/articles/22/9299/2022/, Hunga Tonga 2022: https://www.researchsquare.com/article/rs-1562573/v1)***

We now include all of the above references.

*L28-29: there is "stratosphere" twice, please correct.*

Corrected.

*L30: The Hunga Tonga eruption (the largest in terms of stratospheric aerosol perturbation since Pinatubo 1991) should also be cited here (e.g. https://www.researchsquare.co:1562573/v1 or https://egusphere.copernicus.org/preprints/2022/egusphere-2022-517/)*

Done.

*L30: "low aerosol loading" should maybe be "smaller aerosol perturbations"?*

Changed to "smaller aerosol perturbations".

*L33: why using the past tense ("...this study was...")?*

Corrected.

*L34: please add a reference to explain what a PyroCb is - there are many from the group pf Mike Fromm.*

We revised line 28 to mention PyroCb events and included a reference of Peterson et al., 2018 paper.

*L48: is v2 the newest version of GLOSSAC? In case, please specify*

We revised lines 48 through 50 and now specified the latest version (version 2.2):

"Herein, we describe a cloud screening algorithm for SAGE III/ISS to study the challenges in identifying pure aerosol and aerosol-cloud mixture from SAGE III/ISS observations and their impact on the development of the latest version of GloSSAC (v2.2)."

*L56: why there is such Section 1.1? I would not use subsections of Section 1 (Introduction) and in any case this looks like more a part of Section 2 (Data and Methods)*

Done. Moved this section to Data (section 2) and Methods (section 3).

*L57: "is recently released" should be "was recently released". Please check verb tense throughout the manuscript.*

This has been rewritten. Thanks. (lines 56-57).

*L62: "band of what"? Please correct to "Chappuis ozone absorption band"*

Done. Thanks.

*L66-68 : This is one example of very clumsy sentence. The sentence is not clear, as*

*many others throughout the text. Please try to improve text clarity and the general language throughout the text.*

This has been rewritten (lines 68-72).

*Please add labels a) and b) in Fig 1*

Done. Thanks.

*L77-79: and what about large aerosol perturbations at altitudes larger than 25 km (e.g. for the Hunga Tonga eruption 2022 or the ascending smoke vortices following the Australian fires 2020)? This choice looks like very arbitrary and btw not adapted to asuch events.*

We do not filter out any data above 25 km in the screening. For altitudes above 25 km, negative values mostly occur due to noise and errors in the removal of ozone and molecular scattering and therefore all data above 25 km are retained. This section has been rewritten to make things clearer (lines 70-72).

*L85: Overall, for this issue of negative values and their removal, it sounds like a better understanding for this behaviour is needed, i.e. wrt the inversion algorithm at the basis of SAGEIII/ISS product. Is it a matter of lack of vertical sensitivity? I personally feel that this should be better clarified and such empirical correction is not fully satisfactory as it might screen our a number of points that are actually informative. Why not e.g. resampling the vertical profile at lower vertical resolution? Is there something that can be done by smoothinf the profile with given averaging kernels functions?*

For SAGE-like observations where horizontal homogeneity is a key assumption, where this assumption breaks down is always a data quality issue. Vertical smoothing would likely eliminate negative values but not improve the data quality since the paradigm failure is simply being masked. We have rewritten section to make the issue more clear (lines 76-89).

As far as vertical sensitivity is concerned, the above issue has nothing to do with vertical resolution and vertical averaging will not fix the underlying problem.

*L91: what do you mean with "shape of the distribution"? "Size distribution"? Then, is it a repetition?*

This section is now rewritten (lines 108-123).

*L94-95: please define the "extinction efficiency kernels".*

This section is now rewritten (lines 108-123).

The Figure caption now reads as: (a) shows Mie extinction efficiency kernel ($Q_{(\lambda,r)}$, where Q is extinction efficiency, $\lambda$ is the wavelength, and r is the radii) as a function radius for all SAGE III/ISS wavelengths.

**L98: Please add a refence for "...following large volcanic eruptions". In addition, it depends on the eruption: for Pinatubo and Hunga Tonga, e.g., quite larger average particle sizes were found (see https://egusphere.copernicus.org/preprints/2022/egusphere-2022-517/ for Hunga Tonga)**

We have added a reference now. The sentence now reads as :

"The variations with particle radius in Figure 3b show that at larger particle sizes, the dependence on radius becomes invariant so that above a particle size of about 0.5 µm, all particles have essentially the same 525:1020 extinction ratio. Under most circumstances, particles of this size, or extinction ratios close to 1, are due to the presence of cloud. However, material from intense volcanic eruptions like Mt. Pinatubo or ash, can produce similar ratios (e.g. SPARC, 2006; Legras et al., 2022)."

**Section 2.1: The whole section must be rewritten because it is very unclear. I just stooped reading because I don't understand.**

We have rewritten this section and now use a flow chart to describe the steps used in the method (lines 132-155).

**L103: what do you mean here with "primary aerosol and enhanced aerosol"? "Primary aerosol" is usually used as opposed to "secondary aerosol"**

We change it to "standard aerosol". Thanks.

**L147: "additional" wavelength with respect to what? (SAGEII?)**

This section has been rewritten (lines 158-165).

**L148-149: it sounds like SAGEIII data have been used because there is a negative bias in the 525 nm channel. I don't think that you meant that, thus please rephrase this.**

This sentence has been removed from here. We mention about the negative bias in 525 nm channel in section 2 (line 62) and clarifies the usage of 525 nm extinction coefficient data in GloSSAC in section 4 (lines 427-434).

**L175: "(Thomason and Vernier, 2013)" is always TV13? Why not using this abbreviation?**

Corrected. Thanks.

***Section 2.2.1: What I don't understand here is how the cloud of points for each event is chosen, in particular in terms of time intervals around the date of a specific event, which has an impact on the estimation of the centroid and thus is critical for the overall methodology described in this paper. This choice sounds quite arbitrary and it seems that there are other quite empiric choices through the method description.***

We now use a flow chart to describe the new method. We now discuss the empirical model used for the new method in the supplementary section (S2).

***Section 2.2.2: at this point this reader is lost in the details of the algorithm. The description of the algorithm should really be synthesized and decribed, in terms of the different choices, in a clear and compact manner. I strongly suggest to gather all the different steps of the algorithm in a scheme, a describing flaw-chart figure or something similar***

We now use flow chart to describe the algorithm (section 3.3) and the section has been rewritten.

***L211-213: this sentence is one exemple of the many repetitions throughout the text and thet you should get systematicaly get rid of***

Done. Thanks.

***L214-215: Again, is the choice of using a rigid monthly statistic a good choice for such method? If an event occurs at the beginning or at the end of a month, this is clearly different and the temporal window of such analyses should adapt to this. Why not making averages centered around the actual date of the event?***

We have considered using averages centered around a date using a time window. We think it is interesting, but for this application, we don't see much of a difference in the identification process. Additionally, if we use this method, we do not get enough data points to do any sort of statistics and therefore we used the standard procedure for now.

***L214: They are not exacly at "51 N and 15 S" so use another wording like "at about......"***

Done. We use "∼" before the latitudes. Thanks.

***Section 2.3: I would say that this comparison is only useful if a number of the interesting points (e.g. what is kept with one method and rejected with the other) are studied in more detail. As it stands, Fig 7 is not very useful, it is just a group of 4 clouds of points without any insight about what are the reasons for one method***

*to screen out or keep one point or another. How can we be sure that the new method is better than the old one?*

This section has been rewritten and the figures have been replaced with extinction profiles for the comparison between the two methods (Section 3.4).

*Section 3: why only one case is shown here (Canadian fires 2017)? To be more convincing, I would suggest to show more cases, e.g. in the Supplementary Material of the manuscript*

We are showing additional cases for the comparison using extinction profiles in section S3 of the supplementary.

*Beginning of Section 3: would it be better to introduce GloSSAC in the Methods section? In addition, there are many repetitions here of what discussed in the Introduction, please get rid of these repetitions.*

Done. We now introduce GloSSAC under Methods (section 3.5). Thanks.

*L298-299: please explain why this is an improvement brought by the new method.*

This section is rewritten and discussed in section 4.0.

*Section 3.1: the history of the different versions and the difference amongst them would be much clearer if summarized with a table instead if the lengthy introduction of this section*

This section has been moved to section 3.5 where we describe GloSSAC and the individual data sets used in GloSSAC.

*Sections 3.2 and 3.3: it sounds strange that these sections' titles are almost the same. Why OSIRIS is mention at both? Why not just reorganising in one unique comparison section, merging the two?*

For OSIRIS and CALIOP, we use different methods for the conformance process. We use different sections as the methods used in these sections are different with different data sets.

*Section 3.3: why many methodological aspects of the correlative instruments CALIOP and OSIRIS is described here and not in gthe Methods section? There are many mixed methodological and results information throughout the whole Section 3. Please reorganise your manuscript with a clear structured separation of Methods and Results.*

We have reorganized the sections and all introductory description of GloSSAC is now under Methods

section (section 3.5). The sections for comparison between SAGE III/ISS and OSIRIS/CALIPSO are under section 3.5 in the Methods section (lines 337-416).

***L425-427: this is a clear exemple of sentences that can be easily be made shorter, e.g.: "Figure 13 shows extinction coefficient for September 2017, following Canadian wildfire, for GloSSAC v2.0 and v2.2, as well as their ratios" or something similar. Please be more synthetic throughout the whole text***

Corrected (lines 447-449). Thanks.

***Why "Stratospheric Aerosol Optical Depth" is a Section 4 and not a part of subsection of Section 3 (i.e. a part of the Results section)?***

It is now a section of the results (section 4.2).

***Fig 1: I cannot see any negative extinction value (the x-axis scale goes from 10-7 to 10-1, all ¿0). Is panel b useful at all?***

All the extinction values are plotted as absolute extinction and then we color code negative extinctions as blue and red depending on the error bars on them. The caption is now revised to make it clearer. Thanks.

***Fig 1 caption: "shows how...", is a part of this sentence missing?***

Revised the caption now as it was just a typo. Thanks.

***Tab.1: The pyroconvective cloud activity of Aw started well before 6 January 2020 (I would say 31 December 2019 - visible in OMPS and CALIOP time series since the very beginning of Januiary 2020). Also, what about Hunga Tonga 2022?***

Yes the first Australian pyrocb occurred on 31 December 2019 and the next on 4 January 2020. So, we now use 31 December 2019. Since the focus of the paper was to study the implications of aerosol/cloud categorizations on GloSSAC version 2.2 which was extended until December 2021. That's why we did not include Tonga eruption in the analyses. For SAGE III/ISS data, we reran our code to include data till September 2022, which includes Tonga eruptions as well. However, to extend the GloSSAC data set, we are waiting on CALIPSO level 3 stratospheric aerosol data to be available for the year 2022. We provide several other cases of comparison between TV13 and new methods in the supplementary section (S3) that also includes a profile from Tonga eruption.

***Fig 3, 5, 6 and potentially all figures: please increase size of all in-figure text and labels***

Done. Thanks.

***Fig 13: these panels are very small. Why not a vertical (one column, three lines) orientation?***

This figure has been removed. We use Figure 16 to discuss the differences between the versions.

***Fig 15: if the v2.0 and 2.2 are strictly identical before 2005, why the figure is not just displayed in the period 2005-2021? As it stands, there just is a lot of wasted space and the differences are not really visible as the informative part of the time series just takes a small space in the panels***

Done. It now shows data from 2005 through 2021. Thanks.

***Fig 16: the differences between v2.0 and 2.2 are very difficult to see. As for Fig 15, there is not much interest to see the time series before 2005 as these are identical. In addition, many statistical parameters of the comparisons can be computed (mean bias, RMSE, correlation coefficient, ...) that could help interpreting the differences between v2.0 and 2.2.***

We now include percent differences in the text and point the differences in the figure (lines 473-481) .

**References**

Legras, B., Duchamp, C., Sellitto, P., Podglajen, A., Carboni, E., Siddans, R., Grooß, J.-U., Khaykin, S., and Ploeger, F.: The evolution and dynamics of the Hunga Tonga–Hunga Ha'apai sulfate aerosol plume in the stratosphere, Atmospheric Chemistry and Physics, 22, 14 957–14 970, https://doi.org/10.5194/acp-22-14957-2022, 2022.

SPARC: Assessment of Stratospheric Aerosol Properties (ASAP), Tech. rep., SPARC Report,WCRP-124,WMO/TD-No. 1295,SPARC Report No. 4, 348 pp., 2006.

---

## Author Response (AR2)

**Response to Reviewer #3**

We thank the reviewer for helpful comments. Our responses to the reviewer's specific comments are listed below. The reviewer's concerns are in bold italicized font and our responses are in regular font. The page numbers and line numbers given in our responses below are in reference to the revised version of the manuscript.

*The article has been significantly improved. It represents a significant contribution in the area of stratospheric aerosols.*

**Specific comments**

*Figure 1: I suggest the authors add label for black dots in figure 1 legend (the black dots represent positive extinction coefficients with ¡ 50Describe the horizontal line near the altitude of 12 km in the figure caption. Does this represent tropopause height?*

Done. Thanks.

*Figure 5 and 7: Increase the font size to make it more readable.*

We increased the font size in figure 2, 5, and 7. Thanks.